# Optimal Excitation and Readout of Resonators Used as Wireless Passive Sensors

**DOI:** 10.3390/s24041323

**Published:** 2024-02-18

**Authors:** Leonhard M. Reindl, Taimur Aftab, Gunnar Gidion, Thomas Ostertag, Wei Luo, Stefan Johann Rupitsch

**Affiliations:** 1Laboratory for Electrical Instrumentation and Embedded Systems, Faculty of Engineering, University of Freiburg, 79110 Freiburg, Germany; aftab@imtek.de (T.A.); gunnar.gidion@imtek.uni-freiburg.de (G.G.); stefan.rupitsch@imtek.uni-freiburg.de (S.J.R.); 2RSSI GmbH, Bürgermeister-Graf-Ring 1, 82538 Geretsried, Germany; thomas.ostertag@rssi.de; 3School of Integrated Circuits, Huazhong University of Science and Technology, Wuhan 430074, China; hustluowei@gmail.com

**Keywords:** resonator, transient phenomenon, quality factor, wireless sensor, passive sensor, chipless sensor, analogue sensor, resonant sensor, read-out of resonant sensor

## Abstract

Resonators are passive time-invariant components that do not produce a frequency shift. However, they respond to an excitation signal close to resonance with an oscillation at their natural frequencies with exponentially decreasing amplitudes. If resonators are connected to antennas, they form purely passive sensors that can be read remotely. In this work, we model the external excitation of a resonator with different excitation signals and its subsequent decay characteristics analytically as well as numerically. The analytical modeling explains the properties of the resonator during transient response and decay behavior. The analytical modeling clarifies how natural oscillations are generated in a linear time-invariant system, even if their spectrum was not included in the stimulation spectrum. In addition, it enables the readout signals to be optimized in terms of duration and bandwidth.

## 1. Introduction

Wireless passive sensors are used for applications where conventional battery-based, RFID based, or energy harvesting-based radio technology cannot be operated or only with extensive efforts. The technology is based on the wireless readout of a battery- and IC-free passive sensor node. The instrumentation system consists of a reader unit and a passive sensor node, which are connected via transducers to a wireless link, like a radio, inductive, capacitive or ultrasound link as seen in Figure 1. The reading units are similar to radar systems and various architectures such as time-domain sampling, frequency-domain sampling, and hybrids have been presented in the literature [1,2,3]. Three types of passive sensor nodes are described in the literature: (i) delay lines, (ii) resonators, and (iii) mixer types.

In a delay line, the readout signal is stored for a predefined time interval and then sent back to the reader in one or several pulses [4]. In a resonator type, the readout signal excites a resonator, the oscillation of which decays after the readout signal is switched off [5]. A mixer type either generates a harmonic of the readout signal [6] or mixes two frequencies to an intermediate frequency [7,8]. The mixer type can be combined with a resonator, whereby the mixer demodulates a modulated carrier signal to the modulation frequency for stimulation, which then causes a resonator to oscillate. For reading, the modulation of the interrogation signal is switched off and the oscillating resonator modulates the source resistance of the connected transducer and thus generates a modulated backscatter signal [8].

The separation between the response signal and the readout signal as well as its ambient echoes is implemented in the time domain in the case of delay lines and resonators and in the frequency domain in the mixer types.

The key function of the delay line and the resonator is that they can store the signal as an analog excitation for a period long enough such that all ambient echoes of the readout signal have already decayed. An excellent choice for such an analogous storage is a resonator with a high quality factor *Q*, whose decay time is considerably longer than the power delay profile of the radio channel. When reading out wired or inductively coupled resonators, they will be usually operated in forced oscillation and the narrow-band absorbed power loss at resonance is measured. However, absorption is strongly influenced by the wireless channel and can no longer be evaluated when measuring in the far field. To readout wireless resonators in the far field, an analog storage in the resonant oscillation of the resonators is used [5].

In order to wirelessly poll the information from the far field, the resonator is excited by a readout signal from the reader unit. Electromagnetic, inductive, capacitive, or acoustic channels [9,10] have been used as wireless channel for the far field transmission of the readout and the backscattered decaying signal. While excited, the resonator also oscillates in a forced oscillation in this configuration, but when released, its oscillation will decay with its natural frequency. A part of the energy of the decaying oscillation supplies the response signal, which is scattered back to the reading device and can be recorded and analyzed there. If the resonance frequency is influenced by a physical quantity, this quantity can be determined wirelessly in the reading unit.

In the literature, LC-resonators [11,12], spiral resonators [13], ceramic dielectric resonators [8,14], RF cavity resonators [15,16], coplanar [14,17] and air-filled substrate-integrated waveguide resonators [18], bulk [10], and surface acoustic wave (SAW) resonators [4,19,20,21,22] have been investigated as resonant sensors in passive wireless sensor technology to measure temperature [10,22], pressure [4], torque [19,23], strain [14,15,18,21], mass flow [14], corrosion of reinforced steel [24], pH [12], food quality [25], along with other physical parameters.

Due to the low complexity of the sensor node and due to the operation without any battery and without any electronic circuity, the instrumentation technique is considered to be maintenance-free, robust, and can be operated in harsh environments. The read distance of RFID-based sensor systems is determined by the distance at which power can no longer be extracted from the rectifier, which leads to a threshold value for the read distance. A passive wireless sensor, consisting of a resonator connected to an antenna, is a linear time-invariant system and the readout distance is limited only by the receiver noise, which blocks detection of the response signal from distances beyond the maximum readout distance.

In the scientific literature, the dependences of the resonator response signal on the carrier frequency, the pulse period, the duration of the readout signal and the distance to the reader device, as well as on the modulation spectrum of the physical quantity to be measured have been analyzed numerically and experimentally [26,27,28]. However, an analytical model is still missing.

In this manuscript, we analytically analyze the magnitude of the decaying signal which results from the electrical parameters of the resonator and the temporal waveform of the readout signal. Furthermore, we clear the generation of the signal with the angular natural frequency $\omega_{d}$ out of the readout signal with the frequency ω within the linear time-invariant system. For this purpose, we develop a simple electrical equivalent circuit of the resonator. We calculate the backscattered signals that result from a CW readout signal whose envelope in the time domain corresponds to a rectangular, a trapezoidal, a Tukey window, or which results from a frequency-modulated readout signal with a chirp function. In all analyses, the readout signal with the frequency ω starts at t=0 and ends at t=T. A first decaying signal with frequency $\omega_{d}$ always starts at the beginning of the stimulation at t=0, but in most cases the decaying signal from the end of stimulation at t=T is of interest.

## 2. Modeling the Resonator in a Wireless Readout by Using a Series RLC Circuit Model

The passive wireless sensor node consists of an antenna connected to a resonator. In this analysis, the antenna is simulated by a voltage source $u_{0}$ with real internal resistance $R_{A}$, see Figure 2. When the antenna feeds an incoming signal into the resonator, $R_{A}$ will act as source resistance. On the other hand, if the antenna radiates part of the resonator oscillation, $R_{A}$ will act as sink resistance. In both cases, $R_{A}$ converts power. It is assumed that the quality factor of the antenna is much smaller than the quality factor of the resonator. Therefore, $R_{A}$ is assumed to be constant within the frequency band of interest. RF matching elements are considered part of the resonator. The terminal voltage of the antenna, which is wired to the resonator, is given by $u_{0}-u_{A}$. The resonator is modeled by a serial resonance circuit with capacity *C*, inductance *L*, and dissipative losses modeled by a resistor $R_{D}$. Additional parallel capacitances which often shows up, e.g., in a Butterworth–van-Dyke-equivalent circuit model of a SAW or BAW resonator, are treated as part of the antenna. On the other hand, all ohmic losses in the antenna are taken into account in $R_{D}$. Impedance matching at resonant frequency is assumed meaning $R_{A}=R_{D}$.

If an electromagnetic wave of effective power $P_{in}\left(t\right)$ with the angular frequency ω is picked up by the antenna, an open-circuit voltage $u_{0}\left(t\right)$ with amplitude $U_{0}$ will be created in the internal impedance $R_{A}$, which acts as a source in the circuit:(1)$$u_{0}\left(t\right)=\sqrt{2\cdot P_{in}\left(t\right)\cdot R_{A}}\cdot e^{j\omega t}=U_{0}\cdot e^{j\omega t}.$$

The circuit is a linear time-invariant device, which can be analyzed both in the time domain or the frequency domain. The descriptive differential equation of the system is given by (see Appendix A, Equation (A10)) [28]
(2)$$\frac{d^{2}i\left(t\right)}{{dt}^{2}}+\frac{R_{A}+R_{D}}{L}\frac{di\left(t\right)}{dt}+\frac{i(t)}{LC}=\frac{du_{0}\left(t\right)}{Ldt}.$$By using the following abbreviations:(3)$$2\alpha =\frac{R_{A}+R_{D}}{L},{\omega_{0}}^{2}=\frac{1}{LC}\;,\omega_{d}=\sqrt{{\omega_{0}}^{2}-\alpha^{2}}Q=\frac{X}{R}=\frac{\omega_{0}L}{R_{A}+R_{D}}\;\;\to \alpha =\frac{\omega_{0}}{2Q},$$Equation (Equation 2) can be written in a more general way
(4)$$\frac{d^{2}i\left(t\right)}{{dt}^{2}}+2\alpha \frac{di\left(t\right)}{dt}+{\omega_{0}}^{2}i\left(t\right)=\frac{1}{L}\frac{du_{0}\left(t\right)}{dt}.$$This equation can also be expressed as a function of $u_{A}$ by using the real source impedance $R_{A}$ of the antenna (see Appendix A, Equation (A14))
(5)$$\frac{d^{2}u_{A}\left(t\right)}{{dt}^{2}}+2\alpha \frac{du_{A}\left(t\right)}{dt}+{\omega_{0}}^{2}u_{A}\left(t\right)=\frac{R_{A}}{L}\frac{du_{0}\left(t\right)}{dt}.$$

The vibration characteristics of the system is described by the quality factor *Q* and both the undamped and damped natural angular frequencies $\omega_{0}$ and $\omega_{d}$ respectively. The quality factor is defined by the fraction of the reactance *X* to the resistance *R*. For further analysis, it is helpful to separate the damping constant α into a fraction due to the loading with the antenna $\alpha_{A}$ and due to internal dissipative losses $\alpha_{D}$
(6)$$\alpha_{A}=\frac{R_{A}}{2L},\;\mathrm{and}\;\alpha_{D}=\frac{R_{D}}{2L}.$$

### 2.1. Natural Oscillation with No External Excitation

The general solution $i_{H}\left(t\right)$ of the homogeneous part of Equation (Equation 4) is given by:(7)$$i_{H}\left(t\right)=C_{1}e^{-\alpha t+j\omega_{d}t}+C_{2}e^{-\alpha t-j\omega_{d}t}.$$The actual values of the two complex constants $C_{1}$ and $C_{2}$ of the homogeneous solution result from the boundary conditions. The two solutions of the homogeneous differential equation are called natural oscillations at the natural angular frequencies of the resonator. When the resonator is stimulated, it produces damped free oscillations with the damped natural angular frequencies $\pm \omega_{d}$, whose amplitudes decrease in proportion to $e^{-\alpha t}$, where $0<\alpha <\omega_{0}$ was assumed. The amplitudes drop to $e^{-\pi }\approx 4\%$ after *Q* oscillations. The spectrum of a decaying damped resonator is described by a Lorentz curve.

The currents of the two damped natural oscillations induce voltages across the elements of the circuit. The voltage $u_{AH}$ across $R_{A}$ due to the natural oscillations is given by
(8)$$u_{AH}\left(t\right)=R_{A}\cdot i_{H}\left(t\right).$$Due to the real nature of the source resistance, the antenna’s current and voltage are in phase. The power $P_{out}$, which is taken from the damped natural oscillations in the source resistance of the antenna, is radiated back to the reader unit via the antenna
(9)$$P_{out}\left(t\right)=\frac{1}{2}R\left\{u_{AH}\left(t\right)\cdot {i_{H}}^{*}\left(t\right)\right\}=\frac{{\left|u_{AH}\left(t\right)\right|}^{2}}{2R_{A}}.$$Half of the energy stored in the natural oscillation is radiated back to the reader unit via the antenna due to electrical matching.

### 2.2. Steady State with Sinusoidal Excitation with Constant Amplitude

With a forced periodic excitation by the voltage $u_{0}\left(t\right)$ with constant amplitude $U_{0}$
(10)$$u_{0}\left(t\right)=U_{0}e^{j\omega t},$$
the steady-state current $i_{St}\left(t\right)$ results in (see Appendix A, Equation (A45))
(11)$$i_{St}\left(t\right)=\left(-\frac{\alpha +j\omega_{d}}{\omega -j\alpha +\omega_{d}}+\frac{\alpha -j\omega_{d}}{\omega -j\alpha -\omega_{d}}\right)\cdot \frac{U_{0}}{2\omega_{d}L}\cdot e^{j\omega t}.$$

When the open-circuit voltage $u_{0}$ is generated in the feeding point of the antenna due to picking up of a readout signal, then the steady-state voltage $u_{AG}$ across the source resistance of the antenna at the forced frequency ω will be given by the complex voltage divider
(12)$$u_{AG}=\;u_{0}\cdot \frac{R_{A}}{R_{A}+R_{D}+j\omega L+\frac{1}{j\omega C}}.$$After inserting the abbreviations of Equation (Equation 3) and simplifications (see Equation (A48)), we obtain
(13)$$u_{AG}=u_{0}\cdot \frac{\alpha_{A}}{\omega_{d}}\left(-\frac{a+j\omega_{d}}{\omega -j\alpha +\omega_{d}}+\frac{a-j\omega_{d}}{\omega -j\alpha -\omega_{d}}\right).$$

Equations (Equation 11) and (Equation 13) essentially reflect the same relationship, with Equation (Equation 11) being derived from a solution of the differential equation and Equation (Equation 13) from the steady state, but this is not surprising for linear time-invariant systems. From Equation (Equation 13), we obtain the frequency response $H_{A}\left(\omega \right)$ of the source resistance of the antenna, which is connected in series to the resonator with
(14)$$H_{A}\left(\omega \right)=\frac{u_{A}}{u_{0}}=\frac{\alpha_{A}}{\omega_{d}}\left(-\frac{a+j\omega_{d}}{\omega -j\alpha +\omega_{d}}+\frac{a-j\omega_{d}}{\omega -j\alpha -\omega_{d}}\right).$$

The frequency response can be inverse Fourier transformed to calculate the impulse response of the source impedance of the antenna (see Appendix B, Equation (A65))
(15)$$h_{A}\left(t\right)=\frac{\alpha_{A}}{\omega_{d}}\sigma \left(t\right)\left\{\left(\omega_{d}-j\alpha \right)e^{-\alpha t-j\omega_{d}t}+\left(\omega_{d}+j\alpha \right)e^{-\alpha t+j\omega_{d}t}\right\}.$$For t=0, this results in a value for $h_{A}\left(0\right)$ of
(16)$$h_{A}\left(0\right)=\alpha_{A}=\frac{R_{A}}{R_{A}+R_{D}}\frac{\omega_{0}}{2Q}.$$Figure 3 shows exemplary the frequency response $H_{A}\left(f\right)$ and the corresponding impulse response $h_{A}\left(t\right)$ of an example resonator with center frequency of 1 and a loaded quality factor *Q* of 100. Thereby, electrical matching was assumed, i.e., $R_{A}=R_{D}$.

### 2.3. Boundary Conditions, Transient Phenomenon, and Decay Properties

The voltage $u_{C}$ of the capacitor *C* and the current *i* in the coil *L* correspond to the stored energy in the resonator. Therefore, the values of the voltage $u_{C}$ and of the coil current *i* must be continuous by any change in the externally applied voltage $u_{0}\left(t\right)$. The current *i*, on the other hand, also defines the voltage $u_{A}\left(t\right)$ at the source resistance, which, therefore, must remain continuous with any change in the externally applied voltage. The two damped natural oscillations must compensate any discontinuity in the forced oscillations due to the externally applied voltage $u_{0}\left(t\right)$. Since both the voltages across the resistances and the voltage across the capacitor *C* must remain continuous, any discontinuity in the external open-circuit voltage $u_{0}\left(t\right)$ is entirely applied at the coil. These boundary conditions are required for a direct solution of the differential equation, while they will be automatically fulfilled when solving the differential equation by convolving with the impulse response.

The actual voltages in the circuitry consist of both the generated voltages at the forced frequency ω due to the external voltage $u_{0}$ and the voltages induced due to the currents of the two natural oscillations $\pm \omega_{d}$. The voltage $u_{A}$ across the source resistance of the antenna
(17)$$u_{A}=u_{AG}+u_{AH}$$
corresponds to both the power which is fed into the resonator by the antenna ($u_{AG}$) and the power which is sent back to the reader ($u_{AH}$). In both cases, $R_{A}$ acts as a lossless transformer that converts electromagnetic power into electrical power and back. Since both contributions are included in the impulse response, it is, therefore, sufficient for further analysis to concentrate on $u_{A}$ when calculating the response signal via the impulse response. If we insert Equations (Equation 7) and (Equation 13) into Equation (Equation 17), we will obtain
(18)$$u_{A}\left(t\right)=\frac{\alpha_{A}}{\omega_{d}}\left(-\frac{a+j\omega_{d}}{\omega -j\alpha +\omega_{d}}+\frac{a-j\omega_{d}}{\omega -j\alpha -\omega_{d}}\right)u_{0}\left(t\right)+R_{A}C_{1}e^{-\alpha t+j\omega_{d}t}+R_{A}C_{2}e^{-\alpha t-j\omega_{d}t}.$$

To calculate the response signal by solving the differential equation while taking the boundary conditions into account, it is more advantageous to start from the voltage across the capacitor $u_{C}$. The current resulting from Equation (Equation 18) must be zero as an initial condition for the stored energy of the coil. Integration of this current, therefore, does not result in any further initial condition for the capacitor voltage. The voltage $u_{C}$ results analogously from the externally generated voltage $u_{CG}$ (see (Equation 125)) and from the voltage $u_{CH}$ induced by the two natural oscillations
(19)$$u_{C}=u_{CG}+u_{CH}=u_{0}\cdot \frac{-\omega_{0}^{2}}{\omega^{2}-2j\omega \alpha -\omega_{0}^{2}}+\tilde{C_{1}}e^{-\alpha t+j\omega_{d}t}+\tilde{C_{2}}e^{-\alpha t-j\omega_{d}t},$$
where the two constants $\tilde{C_{1}}$ and $\tilde{C_{2}}$ differ from $C_{1}$ and $C_{2}$ in Equation (Equation 18).

The resonator reacts to every change in the stimulus signal with natural oscillations that ensure the boundary conditions. When these natural oscillations subside, the transition process settles into the steady state. The resonator reacts analogously to the end of stimulation with associated natural oscillations. During the transient process, there is not only a flow of power from the antenna into the resonator but also a return flow from the resonator to the antenna due to the natural oscillations.

At the beginning of the excitation, no current flows. $u_{A}\left(t\right)$ is then very small, and almost the entire open-circuit voltage is fed into the resonator. The current flow only will build up slowly when the resonator begins to oscillate. Since current and voltage change over time during the transient process, the impedance with which the antenna is loaded also changes. During the decay process, the power flows from the resonator into the antenna and is radiated.

The terminal voltage of the antenna, which is wired to the resonator, is given by $u_{0}-u_{AG}$. The power fed into the resonator $P_{fed}\left(t\right)$ is given by
(20)$$P_{fed}\left(t\right)=\left(u_{0}\left(t\right)-u_{AG}\left(t\right)\right)\cdot {i_{St}}^{*}\left(t\right)=\left(u_{0}\left(t\right)-u_{AG}\left(t\right)\right)\cdot \frac{{u_{AG}}^{*}}{R_{A}}.$$

The active power is given by its real part, $R\left\{P_{fed}\left(t\right)\right\}$, and the reactive power by its imaginary part $I\left(P_{fed}\left(t\right)\right)$.

Now that all the necessary formulas are collected to model the resonator connected in series with an antenna, several waveforms can be analyzed that could be used to excite the resonator.

### 2.4. Analytical and Numerical Analysis

To compare the analytical analyses with numerical ones, simulations were carried out using MATLAB [29]. The same formulas, signals, and parameters of the resonator were used in MATLAB as in the analytical calculation. The different window functions used as stimulation signals were implemented in the time domain. For the analytical calculations, the convolution of the stimulation signals with the impulse response of the resonator were calculated analytically; for Section 3, the relevant differential equation was also solved directly, given in Appendix C. To present the results, the parameters of the example resonator, $f_{0}=1$ and Q=100, were inserted into the formulas obtained and the outputs were displayed graphically.

For the MATLAB results, the convolutions were calculated numerically by Fourier transforming the excitation signals, multiplying them by the transfer function of the resonator as given in Equation (Equation 14) and depicted in Figure 3a,b, and then transforming the results back to time domain via IFFT. It is important that the frame data for the numerical simulation are chosen to be sufficiently large in both the time domain and the frequency domain so that aliasing is avoided. In the depicted examples, the resonator has a center frequency of 1 and a quality of 100. To avoid aliasing, the system was modeled with 8192 points and a bandwidth of 10, measured in units of the resonant frequency. This choice ensures sufficient decay of the signals in both the time domain and the frequency domain.

The curves from the numerical calculation lie indistinguishably on the analytically calculated curves in all graphics. To make them visible, the numerically calculated curves were shifted downwards by 0.01 and plotted as red dots. In Figure 3c, the numerically calculated curve was shifted downwards by 1 dB. The simultaneous drawing of the analytically and numerically calculated graphs initially makes it easier to check the analytically calculated formulas.

The numerical simulation can be coded much faster than the analytical calculations, but it only solves this specific example. The analytical formulas, on the other hand, solve the general problem and show the physical processes involved in the transient behavior during excitation and in the generation of the decaying natural oscillations from the excitation spectrum. In addition, they enable optimization of the readout of a resonator.

## 3. Switching the Readout Signal On and Off

The readout signal and, thus, the driving voltage $u_{0}\left(t\right)$ is switched on at t=0 and off att=T:(21)$$u_{0}\left(t\right)=U_{0}e^{j\omega t}\cdot \left\{\begin{array}{ccc} 0 & \mathrm{for}\;t<0 & \mathrm{range}\;I\\ 1 & \mathrm{for}\;0\leq t\leq T & \mathrm{range}\;II\\ 0 & \mathrm{for}\;T<t\; & \mathrm{range}\;III \end{array}\right\}$$The response of the resonator to this stimulation can be analyzed in the time domain either by solving the differential equation (see Appendix C) or by calculating the convolution of the stimulation signal with the impulse response of the resonator (see Appendix D).

### 3.1. Switching On

After the switching on, in range II, we obtain (see Equations (A103) and (A137))
(22)$$\begin{array}{ccc} u_{A}\left(t\right) & = & U_{0}\frac{\alpha_{A}}{\omega_{d}}\left[\left(-\frac{a+j\omega_{d}}{{\omega -j\alpha +\omega }_{d}}+\frac{a-j\omega_{d}}{{\omega -j\alpha -\omega }_{d}}\right)e^{j\omega t}\right.\\ & & \left.+\frac{a+j\omega_{d}}{{\omega -j\alpha +\omega }_{d}}e^{\left(-\alpha {-j\omega }_{d}\right)t}-\frac{a-j\omega_{d}}{{\omega -j\alpha -\omega }_{d}}e^{\left(-\alpha {+j\omega }_{d}\right)t}\right]. \end{array}$$

By inserting Equation (Equation 14), we obtain
(23)$$u_{A}\left(t\right)=U_{0}H_{A}\left(\omega \right)e^{j\omega t}+U_{0}\frac{\alpha_{A}}{\omega_{d}}\left[\frac{a+j\omega_{d}}{{\omega -j\alpha +\omega }_{d}}e^{\left(-\alpha {-j\omega }_{d}\right)t}-\frac{a-j\omega_{d}}{{\omega -j\alpha -\omega }_{d}}e^{\left(-\alpha {+j\omega }_{d}\right)t}\right].$$We immediately obtain the signal for the steady state with sinusoidal excitation, plus two natural oscillations, which together compensate the current in the coil and the voltage in the capacitor. We always obtain both natural oscillations to satisfy the boundary conditions because the excitation occurs with a CW signal, the natural oscillations, however, are damped oscillations, i.e., the driving frequency ω is not an eigenvalue of the differential equation. At the beginning, the sum of the natural oscillations at the frequency ${\;\pm \omega }_{d}$ are at the same amplitude but opposite sign as the forced oscillation at frequency ω. As the natural oscillations gradually decay, the oscillations transition to the stationary state and more and more energy is stored in the resonator.

If the driving frequency is near the angular natural frequency, i.e., ${\omega \approx \omega }_{d}$, and the quality factor *Q* will be high, then the first term of the natural oscillations at the frequency ${-\omega }_{d}$ is by the factor 4Q smaller than the second one at the frequency ${+\omega }_{d}$. Since our driving frequency is +ω, the main part of the induced natural oscillation is at the frequency ${+\omega }_{d}$. However, a small component at the frequency $\;{-\omega }_{d}$ is also required to satisfy the boundary conditions.

If ${\omega \approx \omega }_{d}$ and Q≫1, Equation (Equation 23) can be approximated (see (A141)) to
(24)$$u_{A}\left(t\right)\approx U_{0}H_{A}\left(\omega \right)\left(e^{j\omega t}-e^{\left(-\alpha {+j\omega }_{d}\right)t}\right).$$Since ω is not equal to $\omega_{d}$ in general, a beat might be obtained from the constant forced oscillation and the decaying natural oscillations, which can be seen in Figure 6c. The two terms will add constructively when their difference in angular phase is π
(25)$$\left(\omega -\omega_{d}\right)t=\pi .$$For example, if the resonator is excited at the 3 dB band edge $\omega -\omega_{d}=\frac{1}{2}\omega_{3dB}$, this will result in the optimal excitation length $T_{\omega_{3dB}}$ for the driving voltage of *Q* oscillations, as is chosen in Figure 6b. We see this effect somewhat in Figure 6b, where the response signal, when the excitation signal is switched off, will show an amplitude of 0.36952, i.e., 7% more than we would expect if we only considered their ratio in H(f). However, if the frequency distance is twice the span, e.g., $\omega -\omega_{d}=\omega_{3dB}$, the optimal excitation length for the driving voltage is $\frac{1}{2}Q$ oscillations, as can be seen in Figure 6c. In this case, a stimulation half as long or shorter would result in a significantly higher response signal: 0.27187 for a length of driving voltage of 0.5·Q oscillations when compared to the shown 0.21465 for *Q* oscillations. A length of driving voltage of 0.42·Q oscillations would finally result in a response signal of 0.27877, as can be seen in Figure 9.

This characteristic can also be explained in the frequency domain: the shorter an excitation signal is in the time domain, the broader the main lobe of its spectrum. Therefore, if the carrier frequency of the interrogation signal moves slightly away from the resonance frequency, the position of the resonance frequency will slide downward along the main lobe of the excitation spectrum. In order to pump as much power as possible into the forced oscillation and thus maximize the amplitude of the decay signal, it is, therefore, advantageous to shorten the interrogation signal and, consequently, broaden the main lobe of the spectrum.

At the beginning of the excitation, the current is very small and it is in phase with the applied voltage. As the excitation progresses, the phase shift between the current in the resonator and the voltage applied to the resonator builds up and reaches the value specified in Equation (A26), as can be seen in Figure 4 for three excitation frequencies. The impedance, seen by the source impedance of the antenna, is in the beginning very high, near open end. Figure 5 shows the evolution of the reflection coefficient during the transient phase. With a stimulation at center frequency, the impedance evolves from open to the matched condition along the real axis. With a stimulation frequency next to center frequency, the impedance evolves from open to its steady state value, with frequencies higher than the resonance frequency in the inductive plane and with frequencies lower than resonance in the capacitive one.

Since the impedance is very high at the start of stimulation, only a small fraction of the power offered by the source is injected into the resonator, as can be seen in Figure 8 according to Equation (Equation 20). The active power is shown by the solid black curve and the reactive power by the dashed blue curve. The excitation is performed in Figure 8a at resonance frequency, in Figure 8b at the 3 dB frequency and in Figure 8c at a frequency twice the distance from the resonance. When excited with a resonant frequency, only active power will be transmitted which reaches 100% of the available power after *Q* oscillations. When excited next to the resonance frequency, an increasing amount of reactive power will be transferred and the active power absorption remains well below 100%.

### 3.2. Switching Off

For t>T, we add a second voltage with
(26)$$u_{0}\left(t\right)={-U}_{0}e^{j\omega T}e^{j\omega \left(t-T\right)}.$$This cancels the external voltage $u_{0}$ to zero. The same terms of natural oscillations as in range II show up, however, with alternate signs and time and phase shifted because they now start at t=T. The phase shows the phase shift of ωT of the external voltage between 0 and *T*, and then it increases with $\omega_{d}\left(t-T\right)$. We obtain (see Equations (A109) and (A148)) when the stimulation is switched off
(27)$$u_{A}\left(t\right)=-U_{0}\cdot \frac{\alpha_{A}}{\omega_{d}}\sum_{n=0}^{1}\frac{a+{\left(-1\right)}^{n}j\omega_{d}}{\omega -j\alpha +{\left(-1\right)}^{n}\omega_{d}}\left(e^{j\omega T}-e^{\left(-\alpha -{\left(-1\right)}^{n}j\omega_{d}\right)T}\right)e^{\left(-\alpha -{\left(-1\right)}^{n}j\omega_{d}\right)\left(t-T\right)}.$$

Figure 6 shows in blue the driving voltage and in black the corresponding response signal of the resonator specified in Figure 3 for a driving frequency at resonance frequency at the 3 dB band edge and at twice the 3 dB band edge, which are calculated according to the analytical formulas (Equation 22) and (Equation 27). Additionally, the result of a numerical calculation with MATLAB is shown. For a driving frequency at twice the 3 dB band edge, a length of driving voltage of less than 0.5·Q is preferred, since then the constant forced oscillation and the decaying natural oscillations interfere constructively. The maximum response signal is obtained with 0.42·Q oscillations for the driving signal, which result after switching off the driving signal in a response signal of 0.27877, as can be seen in Figure 7.

To visualize the transient and decay response, Figure 9 shows the real parts of the exciting voltage and the corresponding system response for a resonator with a quality factor of 10.

**Figure 8 sensors-24-01323-f008:**
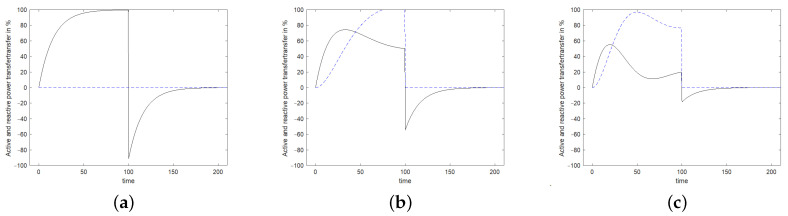
Active power (black solid line) and reactive power (blue dashed line) transferred into the resonator. The excitation is executed in (**a**) at resonance frequency, in (**b**) at the 3 dB frequency and in (**c**) at a frequency twice this distance from the resonance. The stimulating signal starts at t=0 and stops at t=100. After t=100, the power flows from the resonator to the antenna.

The term for n=1 in Equation (Equation 27) is dominant for high Q and $\omega \approx \omega_{d}$. If we ignore the small term for n=0 of the natural oscillations at the frequency ${-\omega }_{d}$ and add the small factor $\frac{j\omega_{d}+\alpha }{{\omega -j\alpha +\omega }_{d}}$ to the remaining natural oscillation, we will obtain for the decay
(28)$$u_{A}\left(t\right)\approx U_{0}H_{A}\left(\omega \right)\left(e^{j\omega T}-e^{-\alpha T}e^{{+j\omega }_{d}T}\right)e^{\left(-\alpha {+j\omega }_{d}\right)\left(t-T\right)}.$$The natural oscillations at the frequency ${\;\omega }_{d}$ consist of two terms. The damping of the larger one starts at t=T, while the damping of the smaller one has already started at t=0. Depending on the phase difference contained in $e^{j\left({\omega -\omega }_{d}\right)T}$, they add up constructively or destructively.

Equation (Equation 28) can also be written as
(29)$$u_{A}\left(t\right)\approx U_{0}H_{A}\left(\omega \right)\left(1-e^{-j\left(\omega -j\alpha -\omega_{d}\right)T}\right)e^{j\omega T}e^{\left(-\alpha {+j\omega }_{d}\right)\left(t-T\right)}.$$If $\left|\alpha +j\left(\omega -\omega_{d}\right)\right|T\ll 1$, i.e., we load only for a short period of time, then both terms will exhibit nearly the same amplitude and mostly cancel each other. In this case, $u_{A}\left(t\right)$ increases with increasing *T*
(30)$$u_{A}\left(t\right)\approx U_{0}H_{A}\left(\omega \right)\left(\alpha +j\left(\omega -\omega_{d}\right)\right)Te^{j\omega T}e^{\left(-\alpha {+j\omega }_{d}\right)\left(t-T\right)}.$$On the other hand, if αT>π, then the second term in the brackets of Equation (Equation 29) can be neglected and the equation simplifies to
(31)$$u_{A,\;cut}\left(t\right)\approx U_{0}H_{A}\left(\omega \right)e^{j\omega T}e^{\left(-\alpha {+j\omega }_{d}\right)\left(t-T\right)}.$$The response signal of the resonator according to Equation (Equation 27) to a rectangular excitation signal consists of two identical response signals, of which the first two occur at the beginning and the other two with a negative sign at the end of the excitation. Each pair has a Lorentz-shaped spectrum around the natural frequency. The time delay in the response signals from the end of the excitation results in a modulation in the frequency range. Depending on the phase of this modulation, the spectral components add up positively or destructively. Since the time interval between the two prompts and the two delayed response signals is the same as between the rising and falling edges of the excitation, their joint spectrum also has the identical zero distribution.

The common spectrum of all response signals and the forced oscillation together is calculated in the frequency domain by multiplying the $\sin\left(\omega \right)/\omega$ function of the spectrum of the excitation signal, which is centered at the carrier frequency, by the Lorentz curve of the spectrum of the resonator. The four response signals alone, therefore, result in a spectrum that corresponds to the “missing” part between the exciting $\sin\left(\omega \right)/\omega$ function and the combined spectrum.

The joint spectral power density of the four response signals was taken from the excitation spectrum. However, the response signals from the end of the excitation alone can contain spectral components that were not included in the excitation spectrum if destructive interference of all Lorentz curves leads to a zero point in the frequency domain. If the response signals from the beginning of the excitation already have decayed at the end of the excitation, a response signal at the natural frequency will show up, even if this frequency was not included in the excitation spectrum.

Figure 10 shows such an example. The carrier of the excitation signal was set to the 6 dB corner frequency of the resonator and the length of the excitation to $Q/f_{0}$. The first zero point of the associated spectrum is, therefore, at the resonance frequency. The top row shows on the left the spectrum of the excitation signal in blue and the Lorentz curve of the resonator in red and on the right the response signal of the resonator to this excitation signal. The bottom row shows on the left the spectrum of the response signal of the resonator after switching off the excitation and on the right the joint spectrum of the response signals from the start and end of the excitation.

If an electromagnetic wave of power $P_{in}$ is picked up by the antenna between t=0 and t=T, an open-source voltage $U_{0}$ will be generated in the resonator circuit (see Equation (Equation 1)). Because the electrical matching is at resonance frequency in the steady state, the loaded voltage over $R_{A}$, $u_{A}$ is half the driving voltage. The power transferred by the antenna from the incoming electromagnetic wave to the resonator is
$$P_{in}=\frac{{u_{A}}^{2}}{R_{A}}=\frac{{U_{0}}^{2}}{4R_{A}}.$$

After switching off the readout signal $P_{in}$, a response signal with power $P_{out}$ is sent back with
(32)$$P_{out}=\frac{{u_{A}}^{2}}{R_{A}}\approx P_{in}{H_{A}}^{2}\left(\omega \right){\left(1-e^{-j\left(\omega -j\alpha -\omega_{d}\right)T}\right)}^{2}e^{j2\omega T}e^{\left(-2\alpha {+j2\omega }_{d}\right)\left(t-T\right)}.$$

Near resonance, i.e., $\omega \approx \omega_{d}$, the term $\left(1-e^{\left(j\omega_{d}-j\omega -\alpha \right)T}\right)$ increases linearly with *T* for αT<1 and approaches 1 for αT>π. If we neglect the constant phase rotation $e^{j2\omega T}$ we will obtain for αT≥π
(33)$$P_{out}\approx P_{in}{H_{A}}^{2}\left(\omega \right)e^{-2\alpha \left(t-T\right)}e^{j2\omega_{d}\left(t-T\right)}.$$

In this case, the power $P_{out}$ during decay starts at the same power level as the picked-up power level $P_{in}$. A longer stimulation phase *T* beyond αT=π does not lead to any further increase in the response signal.

## 4. Increasing and Decreasing the Driving Voltage According to a Trapezoidal Window

A sudden switch on and off of the readout signal and, thus, the driving $u_{A}\left(t\right)$ is not always feasible, e.g., due to legal constraints for RF signals in the ISM bands. In order to reduce the bandwidth of the readout signal, the amplitude can rise and fall according to a Bartlett window or a trapezoidal window, i.e., a Bartlett window with a flat top. For a trapezoidal window, the stimulating signal is
(34)$$u_{0}\left(t\right)=U_{0}e^{j\omega t}\cdot \left\{\begin{array}{ccc} \frac{t}{T_{1}} & \mathrm{for}\;0\leq t\leq T_{1} & \mathrm{range}\;I\\ 1 & \mathrm{for}\;T_{1}\leq t\leq T_{2} & \mathrm{range}\;II\\ \frac{T-t}{T-T_{2}} & \mathrm{for}\;T_{2}\leq t\leq T & \mathrm{range}\;III\\ 0 & \mathrm{for}\;T\leq t & \mathrm{range}\;IV \end{array}\right\}.$$

The response of the system is calculated using the convolution with the impulse response in Appendix F.

### 4.1. Interval with a Linear Increase in the Amplitude of the Stimulating Signal

For the range I when the amplitude of the driving signal is increased linearly in time, the stimulating voltage is given by
(35)$$u_{0}\left(t\right)=U_{0}e^{j\omega t}\cdot \frac{t}{T_{1}}.$$The resonator responds to this stimulation with (see Section F.2, Equation (A163))
(36)$$u_{A}\left(t\right)=\frac{U_{0}}{T_{1}}\cdot \left\{t\cdot H_{A}\left(\omega \right)e^{j\omega t}+\sum_{n=0}^{1}\frac{\alpha_{A}}{\omega_{d}}\frac{\left(\omega_{d}-{\left(-1\right)}^{n}j\alpha \right)}{{\left(\omega -j\alpha +{\left(-1\right)}^{n}\omega_{d}\right)}^{2}}\left[e^{j\omega t}-e^{-\alpha t-{\left(-1\right)}^{n}j\omega_{d}t}\right]\right\}.$$

The resulting amplitude of the driven signal now consists of two parts, one of which is increasing linearly in time and one constant part. The term with the angular natural frequency for n=1 is dominant for Q≫1 and $\omega \approx \omega_{d}$. This dominant term of the angular natural frequency can be expressed as a function of the square of $H_{A}\left(\omega \right)$ (see Equation (A167)). If we ignore all vanishingly small terms, Equation (Equation 36) simplifies to
(37)$$u_{A}\left(t\right)\approx U_{0}\frac{t}{T_{1}}H_{A}\left(\omega \right)e^{j\omega t}+U_{0}\frac{1}{\alpha_{A}T_{1}}\frac{\omega_{d}}{\omega_{d}+j\alpha }H_{A}^{2}\left(\omega \right)\left[e^{-\alpha t+j\omega_{d}t}-e^{j\omega t}\right].$$We obtain a forced term with two parts: one increasing with time and one constant. Due to the continuity of the switch-on process at t=0, the amplitude of the dominant natural oscillation is proportional to the square of $H_{A}\left(\omega \right)$ and, additionally, it is suppressed by $1/{\alpha_{A}T}_{1}$. This natural oscillation is canceled at t=0 by a forced term of the same power and opposite sign.

### 4.2. Interval with a Constant Stimulating Signal

The amplitude of the driving signal is kept constant in range II. The analysis (see Section F.3, Equation (A174)) delivers
(38)$$\begin{array}{ccc} u_{A}\left(t\right) & = & U_{0}\cdot H_{A}\left(\omega \right)e^{j\omega t}+U_{0}\cdot \sum_{n=0}^{1}\frac{\alpha_{A}}{{T_{1}\omega }_{d}}\frac{\left(\omega_{d}-{\left(-1\right)}^{n}j\alpha \right)}{{\left(\omega -j\alpha +{\left(-1\right)}^{n}\omega_{d}\right)}^{2}}\cdot \\ & & \left(-e^{-\alpha t-{\left(-1\right)}^{n}j\omega_{d}t}+e^{j\omega T_{1}}e^{-\alpha \left(t-T_{1}\right)-{\left(-1\right)}^{n}j\omega_{d}\left(t-T_{1}\right)}\right). \end{array}$$

The resulting amplitude of the driven signal is constant and shows the same amplitude as in the case where the amplitude is abruptly switched on. We receive two signals at each of the associated damped angular natural frequencies, one starts to decay at t=0 and the other at $t=T_{1}$. The term of the natural oscillations for n=1 is dominant for Q≫1 and $\omega \approx \omega_{d}$. In this case, we can simplify Equation (Equation 38) to
(39)$$\begin{array}{ccc} u_{A}\left(t\right) & \approx & U_{0}H_{A}\left(\omega \right)e^{j\omega t}\\ & & +U_{0}\frac{\alpha_{A}}{T_{1}\omega_{d}}\frac{\left(\omega_{d}+j\alpha \right)}{{\left(\omega -j\alpha -\omega_{d}\right)}^{2}}\left(-e^{-\alpha t+j\omega_{d}t}+e^{j\omega T_{1}}e^{-\alpha \left(t-T_{1}\right)+j\omega_{d}\left(t-T_{1}\right)}\right). \end{array}$$We obtain two signals oscillating with the angular natural frequency $\omega_{d}$. Depending on their phase difference $e^{j\omega T_{1}}$, they add up constructively or destructively.

### 4.3. Interval with a Linear Decrease in the Amplitude of the Stimulating Signal

The signals in range III, where the amplitude of the driving signal is decreased linearly in time, is similar to the situation in range I, but now with a decreasing driven signal. The slope of the decreasing is sometimes set faster than the increasing amplitude slope, therefore, a different slope was chosen. The response signals of the systems are (see Section F.4, Equation (A183)):(40)$$\begin{array}{ccc} u_{A}\left(t\right) & = & U_{0}\frac{T-t}{T-T_{2}}H_{A}\left(\omega \right)e^{j\omega t}+U_{0}\frac{\alpha_{A}}{\omega_{d}}\sum_{n=0}^{1}\frac{\left(\omega_{d}-{\left(-1\right)}^{n}j\alpha \right)}{{\left(\omega -j\alpha +{\left(-1\right)}^{n}\omega_{d}\right)}^{2}}\cdot \\ & & \;\left[\frac{1}{T_{1}}\left(e^{j\omega T_{1}}e^{-\alpha \left(t-T_{1}\right)-{\left(-1\right)}^{n}j\omega_{d}\left(t-T_{1}\right)}-e^{-\alpha t-{\left(-1\right)}^{n}j\omega_{d}t}\right)+\right.\\ & & \;\left.+\frac{1}{\left(T-T_{2}\right)}\left(e^{j\omega T_{2}}e^{-\alpha \left(t-T_{2}\right)-{\left(-1\right)}^{n}j\omega_{d}\left(t-T_{2}\right)}-e^{j\omega t}\right)\right] \end{array}$$

We have a linearly decreasing driven signal and a constant driven signal, which is exactly compensated at ${t=T}_{2}$ by a damped natural oscillation. At the switching points t=0 and ${t=T}_{1}$, we obtain two natural oscillations, one at $-\omega_{d}$ and one at $+\omega_{d}$, which start at *t* = 0 and *t* = $T_{1}$ and then die out. The natural oscillation again can be expanded as a function of the square of $H_{A}\left(\omega \right)$. If we keep only the dominating terms for Q≫1 and $\omega \approx \omega_{d}$, Equation (Equation 40) simplifies to (see Equation (A186))
(41)$$\begin{array}{ccc} u_{A}\left(t\right) & \approx & U_{0}\frac{T-t}{T-T_{2}}H_{A}\left(\omega \right)e^{j\omega t}-U_{0}\frac{\omega_{d}}{\left(\omega_{d}+j\alpha \right)\alpha_{A}}H_{A}^{2}\left(\omega \right)\cdot \\ & & \left[-\frac{1}{T_{1}}e^{-\alpha t+j\omega_{d}t}+\frac{1}{T_{1}}e^{j\omega T_{1}}e^{-\alpha \left(t-T_{1}\right)+j\omega_{d}\left(t-T_{1}\right)}+\right.\\ & & \left.\frac{1}{\left(T-T_{2}\right)}e^{j\omega T_{2}}e^{-\alpha \left(t-T_{2}\right)+j\omega_{d}\left(t-T_{2}\right)}-\frac{1}{\left(T-T_{2}\right)}e^{j\omega t}\right]. \end{array}$$Of most practical interest are the remaining signals after the driving has stopped.

### 4.4. Switching the Stimulating Signal Off

After the end of the stimulation signal, the resonator oscillates on its natural oscillations (see Section F.5, Equation (A191))
(42)$$\begin{array}{ccc} u_{A}\left(t\right) & = & U_{0}\frac{\alpha_{A}}{\omega_{d}}\sum_{n=0}^{1}\frac{\left(\omega_{d}+{\left(-1\right)}^{n}j\alpha \right)}{{\left(\omega -j\alpha -{\left(-1\right)}^{n}\omega_{d}\right)}^{2}}\left[-\frac{1}{T_{1}}e^{-\alpha t+{\left(-1\right)}^{n}j\omega_{d}t}+\frac{1}{T_{1}}e^{j\omega T_{1}}e^{-\alpha \left(t-T_{1}\right)+{\left(-1\right)}^{n}j\omega_{d}\left(t-T_{1}\right)}+\right.\\ & & \left.\frac{1}{\left(T-T_{2}\right)}e^{j\omega T_{2}}e^{-\alpha \left(t-T_{2}\right)+{\left(-1\right)}^{n}j\omega_{d}\left(t-T_{2}\right)}-\frac{1}{\left(T-T_{2}\right)}e^{j\omega T}e^{-\alpha \left(t-T\right)+{\left(-1\right)}^{n}j\omega_{d}\left(t-T\right)}\right]. \end{array}$$

The linearly decreasing driven part of the signals vanished after switching off the driving signal at t=T. The constant driven signals, however, transformed into a decaying natural oscillation at t=T. Thus, at each switching point we obtain two natural oscillations, one at $-\omega_{d}$ and one at $+\omega_{d}$, which start at *t* = 0, *t* = $T_{1}$, *t* = $T_{2}$ and *t* = *T*, and then die out exponentially. Depending on their sign and relative phase shift 0, $e^{j\omega T_{1}}$, $e^{j\omega T_{2}}$, $e^{j\omega T}$ they add up constructively or destructively.

Figure 11 shows in blue the driving voltage and in black the corresponding response signal of the resonator specified in Figure 3 for a driving frequency at resonance frequency, at the 3 dB band edge and at twice the 3 dB band edge, which are calculated according to the analytical formulas (Equation 36), (Equation 38) and (Equation 42). Additionally, the result of a numerical calculation with MATLAB is shown.

For a driving frequency at the 3 dB band edge, or at twice the 3 dB band edge, shorter stimulations with fewer than Q oscillations lead to higher response signals. Figure 12 shows the response for shorter stimulations at frequencies at the 3 dB band edge and at twice of it.

Surprisingly, the length $\left(T_{2}-T_{1}\right)$ of the constant stimulation does not seem to play a direct role in the Equation (Equation 42), but only the rate of rise and fall of the stimulation voltage. However, if the length $\left(T_{2}-T_{1}\right)$ is chosen too short, the destructive interference that occurs between the individual terms of Equation (Equation 42) will lead to a drastic reduction in the overall response. Furthermore, it is astonishing that the denominators of the individual parts contain the rise and fall times $T_{1}$ and$\;T-T_{2}$. Since these can be selected freely, it seems possible that a stronger decay signal might be generated by a clever choice of the rise and fall rates, compared to the case with hard switching on and off of the driving signal. An analysis shows that any possible increase in signal strength by shortening the rise and fall rates is also deteriorated by a destructive interference of the associated terms of natural oscillation.

The term with the first large square bracket will be the dominant one of Equation (Equation 42) when Q≫1 and $\omega \approx \omega_{d}$. When the time for constant stimulation is chosen long enough to fully load the resonator, i.e., $\left(T_{2}-T_{1}\right)\cdot \alpha >\pi$, then the decay terms starting at t=0 and $t=T_{1}$ can also be ignored. In this case, Equation (Equation 42) simplifies to
(43)$$\begin{array}{ccc} u_{A}\left(t\right) & \approx & U_{0}\frac{\alpha_{A}}{\omega_{d}}\frac{-\left(\omega_{d}+j\alpha \right)}{{\left(\omega -j\alpha -\omega_{d}\right)}^{2}}\cdot \left[\frac{1}{\left(T-T_{2}\right)}e^{j\omega T}e^{-\alpha \left(t-T\right)+j\omega_{d}\left(t-T\right)}\right.\\ & & \left.-\frac{1}{\left(T-T_{2}\right)}e^{j\omega T_{2}}e^{-\alpha \left(t-T_{2}\right)+j\omega_{d}\left(t-T_{2}\right)}\right]. \end{array}$$
(44)$$u_{A}\left(t\right)\approx U_{0}\frac{\alpha_{A}}{\omega_{d}}\frac{-\left(\omega_{d}+j\alpha \right)}{{\left(T-T_{2}\right)\left(\omega -j\alpha -\omega_{d}\right)}^{2}}\left[1-e^{-j\left(\omega -\omega_{d}-j\alpha \right)\left(T-T_{2}\right)}\right]e^{j\omega T}e^{-\alpha \left(t-T\right)+j\omega_{d}\left(t-T\right)}$$The two terms might add constructively if their phase difference is π. However, in this case either the frequency response $H_{A}\left(\omega \right)$ will be very small (see Section F.5, (A198)), i.e., the driving frequency ω is not within the resonance, or the second term in Equation (Equation 44) will already have nearly faded away. The two terms in Equation (Equation 44) will, therefore, always add destructively. Increasing $\left(T-T_{2}\right)$ so that the second term in the square bracket can be neglected does not help either, as it also reduces the prefactor of the bracket. On the other hand, if the decrease time $\left(T-T_{2}\right)$ is chosen fast enough to ensure
$$\left|\omega -\omega_{d}-j\alpha \right|\left(T-T_{2}\right)\ll 1,$$
then we can approximate the exponential function up to the quadratic term and obtain (see Section F.5, (A203))
(45)$$\begin{array}{ccc} u_{A}\left(t\right) & \approx & U_{0}H_{A}\left(\omega \right)e^{j\omega T}e^{-\alpha \left(t-T\right)+j\omega_{d}\left(t-T\right)}\cdot \\ & & \left[1-\frac{1}{2}\left(\alpha -j\omega_{d}+j\omega \right)\left(T-T_{2}\right)+\frac{1}{6}{\left(\alpha -j\omega_{d}+j\omega \right)}^{2}{\left(T-T_{2}\right)}^{2}\right]. \end{array}$$A comparison of the decay signal in the case of applying a trapezoidal window on the readout signal $u_{A,\;Tapezoidal}\left(t\right)$ to the decay signal in the case of hard switch on and off of the readout signal $u_{A,\;cut}\left(t\right)$ gives
(46)$$\begin{array}{ccc} u_{A,\;Tapezoidal}\left(t\right) & \approx & u_{A,\;cut}\left(t\right)\cdot \left[1-\frac{1}{2}\left(\alpha -j\omega_{d}+j\omega \right)\left(T-T_{2}\right)\right.\\ & & \left.+\frac{1}{6}{\left(\alpha -j\omega_{d}+j\omega \right)}^{2}{\left(T-T_{2}\right)}^{2}\right]. \end{array}$$

That is to say, we obtain a decay signal with nearly the same strength as in the case with hard switch on and off, if the amplitude decrease time $\left(T-T_{2}\right)$ is short enough.

### 4.5. Increasing and Decreasing the Driving Voltage According to a Bartlett Window

The last sections showed that the level of the decay signal depends crucially on the length of time of the constant excitation and on a sufficiently fast decay of the excitation signal. How does the situation change if we skip the constant loading and use a driving signal according to a Bartlett window. The corresponding loading signal is given by
(47)$$u_{0}\left(t\right)=U_{0}e^{j\omega t}\cdot \left\{\begin{array}{ccc} \frac{2t}{T} & \mathrm{for}\;0\leq t\leq \frac{T}{2} & \mathrm{range}\;I\\ \frac{2\left(T-t\right)}{T} & \mathrm{for}\;\frac{T}{2}\leq t\leq T & \mathrm{range}\;II\\ 0 & \mathrm{for}\;T\leq t & \mathrm{range}\;II \end{array}\right\}$$

The response of the system can be calculated by using the results for a trapezoidal loading signal and adopting the times $T_{1}=T_{2}=T/2$ (see Section F.6). Substituting $T_{1}=T/2$ into (Equation 36) gives the signals during the increasing of the stimulation
(48)$$u_{A}\left(t\right)=U_{0}\frac{2t}{T}H_{A}\left(\omega \right)e^{j\omega t}+U_{0}\frac{{2\alpha }_{A}}{T\omega_{d}}\sum_{n=0}^{1}\frac{\left(\omega_{d}-{\left(-1\right)}^{n}j\alpha \right)}{{\left(\omega -j\alpha +{\left(-1\right)}^{n}\omega_{d}\right)}^{2}}\left[e^{j\omega t}-e^{-\alpha t-{\left(-1\right)}^{n}j\omega_{d}t}\right].$$For the decreasing part, the substitution of $T_{1}=T_{2}=T/2$ in Equation (Equation 40) results in
(49)$$\begin{array}{ccc} u_{A}\left(t\right) & = & 2U_{0}\frac{T-t}{T}H_{A}\left(\omega \right)e^{j\omega t}+U_{0}\frac{2}{T}\frac{\alpha_{A}}{\omega_{d}}\sum_{n=0}^{1}\frac{\left(\omega_{d}-{\left(-1\right)}^{n}j\alpha \right)}{{\left(\omega -j\alpha +{\left(-1\right)}^{n}\omega_{d}\right)}^{2}}\cdot \\ & & \left[2e^{j\omega \frac{T}{2}}e^{-\alpha \left(t-\frac{T}{2}\right)-{\left(-1\right)}^{n}j\omega_{d}\left(t-\frac{T}{2}\right)}-e^{-\alpha t-{\left(-1\right)}^{n}j\omega_{d}t}-e^{j\omega t}\right]. \end{array}$$And, finally, the decaying natural oscillation for t>T, when stimulation is completed, results by the substitution of $T_{1}=T_{2}=T/2$ into the Equation (Equation 42)
(50)$$\begin{array}{ccc} u_{A}\left(t\right) & = & U_{0}\frac{2}{T}\frac{\alpha_{A}}{\omega_{d}}\sum_{n=0}^{1}\frac{\left(\omega_{d}-{\left(-1\right)}^{n}j\alpha \right)}{{\left(\omega -j\alpha +{\left(-1\right)}^{n}\omega_{d}\right)}^{2}}\left[-e^{-\alpha t-{\left(-1\right)}^{n}j\omega_{d}t}+\right.\\ & & \left.\;+2e^{j\omega \frac{T}{2}}e^{-\alpha \left(t-\frac{T}{2}\right)-{\left(-1\right)}^{n}j\omega_{d}\left(t-\frac{T}{2}\right)}-e^{j\omega T}e^{-\alpha \left(t-T\right)-{\left(-1\right)}^{n}j\omega_{d}\left(t-T\right)}\right]. \end{array}$$For the Equations (48)–(50) approximations for Q≫1 and $\omega \approx \omega_{d}$ can be estimated, which are given in Section F.6, Equations (A208) and (A212). Neglecting the vanishing small second term in (50) leads to (see also Equation A215)
(51)$$u_{A}\left(t\right)\approx U_{0}\frac{2}{T\left(\omega -j\alpha -\omega_{d}\right)}H_{A}\left(\omega \right){\left(1-e^{-j\left(\omega -j\alpha -\omega_{d}\right)\frac{T}{2}}\right)}^{2}e^{j\omega T}e^{-\alpha \left(t-T\right)+j\omega_{d}\left(t-T\right)}$$
(52)$$u_{A,\;Bartlett}\left(t\right)\approx u_{A,\;cut}\left(t\right)\frac{2}{T\left(\omega -j\alpha -\omega_{d}\right)}{\left(1-e^{-j\left(\omega -j\alpha -\omega_{d}\right)\frac{T}{2}}\right)}^{2}.$$$u_{A,\;Bartlett}\left(t\right)$ increases linear with *T* for small *T* (see Equation (A220)), and reaches a maximum at Tα=2.5 (see Equation (A224)) with
$$Tf=T\frac{\omega }{2\pi }=\frac{2.5}{\pi }Q=0,8Q.$$
(53)$${\left.u_{A,\;Bartlett}\left(t\right)\right|}_{T\alpha =2.5}\approx 0.41u_{A,\;cut}\left(t\right)$$Figure 13 shows, in blue, the driving voltage and, in black, the corresponding response signal of the resonator specified in Figure 3 for a driving signal which is weighted in the time domain with a triangle function (Bartlett window). The driving frequency was set to resonance frequency at the 3 dB band edge and at twice the 3 dB band edge. The lengths of the Bartlett windows were optimized to maximum response signal at the time when the driving signal was switched off.

## 5. Weighting the Driving Voltage by Using a Tukey Window

The driving voltage $u_{0}\left(t\right)$ now rises and decreases according to a modified Tukey window with a flat top and optional different raise and fall-off rates
(54)$$u_{0}\left(t\right)=U_{0}e^{j\omega t}\cdot \left\{\begin{array}{ccc} \;\frac{1}{2}\cdot \left(1-\cos\frac{\pi t}{T_{1}}\;\right) & \mathrm{for}\;0\leq t\leq T_{1}\;\; & \mathrm{range}\;I\;\;\;\;\\ 1 & \mathrm{for}\;T_{1}\leq t\leq T_{2} & \mathrm{range}\;II\;\;\\ \frac{1}{2}\cdot \left(1-\cos\frac{\pi \left(T-t\right)}{\left(T-T_{2}\right)}\;\right) & \mathrm{for}\;T_{2}\leq t\leq T\; & \mathrm{range}\;III\\ 0 & \mathrm{for}\;T\leq t\;\;\;\;\;\;\;\;\;\; & \;\mathrm{range}\;IV\;\; \end{array}\right\}$$

In the rising part of the Tukey weighting, we receive the following signal from the resonator as a response (see Appendices Appendix G and Section G.2, Equation (A245))
(55)$$\begin{array}{ccc} u_{A}\left(t\right) & = & \frac{1}{2}U_{0}\frac{\alpha_{A}}{\omega_{d}}\sum_{n=0}^{1}\frac{\left(\omega_{d}-{\left(-1\right)}^{n}j\alpha \right)}{\left(\alpha +{\left(-1\right)}^{n}j\omega_{d}+j\omega \right)\left[{\left(\alpha +{\left(-1\right)}^{n}j\omega_{d}+j\omega \right)}^{2}+{\left(\frac{\pi }{T_{1}}\right)}^{2}\right]}\cdot \\ & & \;\left[-{\left(\frac{\pi }{T_{1}}\right)}^{2}e^{-\alpha t-{\left(-1\right)}^{n}j\omega_{d}t}+\left\{{\left(\frac{\pi }{T_{1}}\right)}^{2}+{\left(\alpha +{\left(-1\right)}^{n}j\omega_{d}+j\omega \right)}^{2}\left[1-\cos\left(\pi \frac{t}{T_{1}}\right)\;\right]\right.\right.\\ & & \left.\left.\;\;\;-\frac{\pi }{T_{1}}\left(\alpha +{\left(-1\right)}^{n}j\omega_{d}+j\omega \right)\sin\left(\pi \frac{t}{T_{1}}\right)\;\right\}e^{j\omega t}\right]. \end{array}$$

For the portion with constant charging of the resonator (range II), we obtain (see Section G.3, Equation (A250))
(56)$$\begin{array}{ccc} u_{A}\left(t\right) & = & \frac{1}{2}U_{0}\frac{\alpha_{A}}{\omega_{d}}\sum_{n=0}^{1}\frac{\left(\omega_{d}-{\left(-1\right)}^{n}j\alpha \right)}{\left(\alpha +{\left(-1\right)}^{n}j\omega_{d}+j\omega \right)}\cdot \left[2e^{j\omega t}-\frac{{\left(\frac{\pi }{T_{1}}\right)}^{2}}{{\left(\alpha +{\left(-1\right)}^{n}j\omega_{d}+j\omega \right)}^{2}+{\left(\frac{\pi }{T_{1}}\right)}^{2}}\cdot \right.\\ & & \left.\left(e^{-\alpha t-{\left(-1\right)}^{n}j\omega_{d}t}+e^{j\omega T_{1}}e^{-\left(\alpha +{\left(-1\right)}^{n}j\omega_{d}\right)\left(t-T_{1}\right)}\right)\right]. \end{array}$$

The solution for the cosine-shaped decrease in the amplitude of the stimulation signal results in (see Section G.4, Equation (A255))
(57)$$\begin{array}{ccc} u_{A}\left(t\right) & = & \frac{1}{2}U_{0}\frac{\alpha_{A}}{\omega_{d}}\sum_{n=0}^{1}\frac{\left(\omega_{d}-{\left(-1\right)}^{n}j\alpha \right)}{\left(\alpha +{\left(-1\right)}^{n}j\omega_{d}+j\omega \right)}\cdot \\ & & \left[-\frac{{\left(\frac{\pi }{T_{1}}\right)}^{2}}{{\left(\alpha +{\left(-1\right)}^{n}j\omega_{d}+j\omega \right)}^{2}+{\left(\frac{\pi }{T_{1}}\right)}^{2}}\left(e^{-\alpha t-{\left(-1\right)}^{n}j\omega_{d}t}+e^{j\omega T_{1}}e^{-\left(\alpha +{\left(-1\right)}^{n}j\omega_{d}\right)\left(t-T_{1}\right)}\right)\right.\\ & & \;+\frac{1}{{\left(\alpha +{\left(-1\right)}^{n}j\omega_{d}+j\omega \right)}^{2}+{\left(\frac{\pi }{\left(T-T_{2}\right)}\right)}^{2}}\left\{{\left(\frac{\pi }{\left(T-T_{2}\right)}\right)}^{2}e^{j\omega T_{2}}e^{-\left(\alpha +{\left(-1\right)}^{n}j\omega_{d}\right)\left(t-T_{2}\right)}\right.\\ & & \;\;+\left[{\left(\frac{\pi }{\left(T-T_{2}\right)}\right)}^{2}+{\left(\alpha +{\left(-1\right)}^{n}j\omega_{d}+j\omega \right)}^{2}\left[1-\cos\left(\pi \frac{\left(T-t\right)}{\left(T-T_{2}\right)}\right)\;\right]\right.\\ & & \left.\left.\left.\;\;\;+\frac{\pi }{\left(T-T_{2}\right)}\left(\alpha +{\left(-1\right)}^{n}j\omega_{d}+j\omega \right)\sin\left(\pi \frac{\left(T-t\right)}{\left(T-T_{2}\right)}\right)\;\right]e^{j\omega t}\right\}\right]. \end{array}$$

After the end of the stimulation signal, the resonator oscillates with its decaying natural oscillations (see Section G.5, Equation (A260))
(58)$$\begin{array}{ccc} u_{A}\left(t\right) & = & \frac{1}{2}U_{0}\frac{\alpha_{A}}{\omega_{d}}\sum_{n=0}^{1}\frac{\left(\omega_{d}-{\left(-1\right)}^{n}j\alpha \right)}{\left(\alpha +{\left(-1\right)}^{n}j\omega_{d}+j\omega \right)}\cdot \\ & & \left[-\frac{{\left(\frac{\pi }{T_{1}}\right)}^{2}}{{\left(\alpha +{\left(-1\right)}^{n}j\omega_{d}+j\omega \right)}^{2}+{\left(\frac{\pi }{T_{1}}\right)}^{2}}\left(e^{-\alpha t-{\left(-1\right)}^{n}j\omega_{d}t}+e^{j\omega T_{1}}e^{-\left(\alpha +{\left(-1\right)}^{n}j\omega_{d}\right)\left(t-T_{1}\right)}\right)+\right.\\ & & +\frac{{\left(\frac{\pi }{\left(T-T_{2}\right)}\right)}^{2}}{{\left(\alpha +{\left(-1\right)}^{n}j\omega_{d}+j\omega \right)}^{2}+{\left(\frac{\pi }{\left(T-T_{2}\right)}\right)}^{2}}\cdot \\ & & \left.\left(e^{j\omega T_{2}}e^{-\left(\alpha +{\left(-1\right)}^{n}j\omega_{d}\right)\left(t-T_{2}\right)}+e^{j\omega T}e^{-\left(\alpha +{\left(-1\right)}^{n}j\omega_{d}\right)\left(t-T\right)}\right)\right]. \end{array}$$

Figure 14 shows in blue the driving voltage and in black the corresponding response signal of the resonator specified in Figure 3 for a driving frequency at resonance frequency, at the 3 dB band edge and at twice the 3 dB band edge, which are calculated according to the analytical Equations (55)–(58). Additionally, the result of a numerical calculation with MATLAB is shown. Figure 15 shows that for stimulation signals at the 3 dB band edge and at twice of that, the response signal of the resonator is increased also for Tukey-weighted stimulation signals, when the stimulation interval is shortened.

At each switching point, we obtain two natural oscillations that start at *t* = 0, *t* = $T_{1}$, *t* = $T_{2}$, and *t* = *T* and then die out. The four natural oscillations for n=1 in Equation (58) are dominant for resonators with high Q and a stimulation near resonance. If $\alpha \left(T_{2}-T_{1}\right)>1$, the natural oscillations starting at t=0 and $t=T_{1}$ can also be ignored, and Equation (58) can be simplified to
(59)$$\begin{array}{ccc} u_{A}\left(t\right) & \approx & \frac{1}{2}U_{0}\frac{\alpha_{A}}{\omega_{d}}\frac{\left(\omega_{d}-j\alpha \right)}{\left(\alpha +j\omega_{d}+j\omega \right)}\cdot \frac{{\left(\frac{\pi }{\left(T-T_{2}\right)}\right)}^{2}}{{\left(\alpha -j\omega_{d}+j\omega \right)}^{2}+{\left(\frac{\pi }{\left(T-T_{2}\right)}\right)}^{2}}\cdot \\ & & \left(e^{j\omega T_{2}}e^{-\left(\alpha -j\omega_{d}\right)\left(t-T_{2}\right)}+e^{j\omega T}e^{-\left(\alpha -j\omega_{d}\right)\left(t-T\right)}\right). \end{array}$$

A good approximation for this result is (see Section G.5, Equation (A264))
(60)$$\begin{array}{ccc} u_{A}\left(t\right) & \approx & \frac{1}{2}U_{0}H_{A}\left(\omega \right)\frac{\pi^{2}\left(1+e^{-j\left(\omega -j\alpha -\omega_{d}\right)\left(T-T_{2}\right)}\right)}{{\left(T-T_{2}\right)}^{2}{\left(\alpha -j\omega_{d}+j\omega \right)}^{2}+\pi^{2}}\cdot e^{j\omega T}e^{-\left(\alpha -j\omega_{d}\right)\left(t-T\right)}. \end{array}$$

If we expand the fraction and the exponential function in the brackets, we obtain (see Section G.5, Equation (A268))
(61)$$\begin{array}{ccc} u_{A,\;Tukey}\left(t\right) & \approx & U_{0}H_{A}\left(\omega \right)\left(1-\frac{1}{2}\left(\alpha -j\omega_{d}+j\omega \right)\left(T-T_{2}\right)+\right.\\ & & \left.+\left(\frac{1}{4}-\frac{1}{\pi^{2}}\right){\left(\alpha -j\omega_{d}+j\omega \right)}^{2}{\left(T-T_{2}\right)}^{2}\right)e^{j\omega T}e^{-\left(\alpha -j\omega_{d}\right)\left(t-T\right)}. \end{array}$$

A comparison of the decay signal in the case of applying a Tukey window on the readout signal $u_{A,\;Tukey}\left(t\right)$ to the decay signal in the case of hard switch on and off of the readout signal $u_{A,\;cut}\left(t\right)$ gives
(62)$$\begin{array}{ccc} u_{A,\;Tukey}\left(t\right) & \approx & u_{A,\;cut}\left(t\right)\cdot \left[1-\frac{1}{2}\left(\alpha -j\omega_{d}+j\omega \right)\left(T-T_{2}\right)\right.\\ & & \left.+\left(\frac{1}{4}-\frac{1}{\pi^{2}}\right){\left(\alpha -j\omega_{d}+j\omega \right)}^{2}{\left(T-T_{2}\right)}^{2}\right]. \end{array}$$Tukey-windowed excitation signals require a lower bandwidth than a square window signal. However, their response signals also reach a lower level at the time the readout signal stops because the additional excitation in the falling edge does not fully compensate for the exponential decay of the response signal. Figure 16 shows this decrease in the bandwidth of the stimulation signal together with the resulting additional decrease in the response signal. Here, cosine-weighted end sections were attached to a constant stimulation signal at both ends. The stimulation frequencies in the graphs are $f_{0}$ in the left, $0.995\cdot f_{0}$ in the center and $0.99\cdot f_{0}$ at the right. The lengths of the constant stimulation signals are set to $Q/f_{0}$ for the left graph, $0.75\cdot Q/f_{0}$ in the center and $0.3\cdot Q/f_{0}$ in the right one. The spectrum of the stimulation signals contains many small side lobes. Depending on whether a side lobe contributes to or falls below the −50 dB bandwidth, the resulting bandwidth jumps up or down.

### Weighting the Driving Voltage by Using Hann Window

In the case of a Hann window, the driving voltage $u_{0}\left(t\right)$ rises and decreases according to a cosine window
(63)$$u_{0}\left(t\right)=U_{0}e^{j\omega t}\cdot \left\{\begin{array}{ccc} \frac{1}{2}\cdot \left(1-\cos\frac{2\pi t}{T}\;\right) & \mathrm{for}\;0\leq t\leq T & \mathrm{range}\;I\\ 0 & \mathrm{for}\;T\leq t & \mathrm{range}\;IV \end{array}\right\}.$$

We obtain for t>T (see Section G.6, Equation (A272))
(64)$$\begin{array}{ccc} u_{A}\left(t\right) & = & \frac{1}{2}U_{0}\frac{\alpha_{A}}{\omega_{d}}\sum_{n=0}^{1}\frac{{\left(\frac{2\pi }{T}\right)}^{2}}{{\left(\alpha +{\left(-1\right)}^{n}j\omega_{d}+j\omega \right)}^{2}+{\left(\frac{2\pi }{T}\right)}^{2}}\frac{\left(\omega_{d}-{\left(-1\right)}^{n}j\alpha \right)}{\left(\alpha +{\left(-1\right)}^{n}j\omega_{d}+j\omega \right)}\cdot \\ & & \;\;\;\;\left[-e^{-\alpha t-{\left(-1\right)}^{n}j\omega_{d}t}+e^{j\omega T}e^{-\left(\alpha +{\left(-1\right)}^{n}j\omega_{d}\right)\left(t-T\right)}\right]. \end{array}$$

We obtain two decay signals, one which is triggered by the start of the loading, with an amplitude proportional to $e^{-\alpha t}$, and one which is triggered by the end of the loading, with an amplitude proportional to $e^{-\alpha \left(t-T\right)}$. For resonators with high Q and a stimulation near resonance, we can skip the small terms for n=0 and add the small term $\frac{\omega_{d}-ja}{\alpha +j\omega +j\omega_{d}}$ and obtain (Equation (A274))
(65)$$u_{A}\left(t\right)\approx U_{0}\frac{1}{2}H_{A}\left(\omega \right)\left(\frac{{\left(\frac{2\pi }{T}\right)}^{2}}{{\left(\alpha -j\omega_{d}+j\omega \right)}^{2}+{\left(\frac{2\pi }{T}\right)}^{2}}\right)\left(1-e^{-j\left(\omega -j\alpha -\omega_{d}\right)T}\right)e^{j\omega T}e^{-\left(\alpha -j\omega_{d}\right)\left(t-T\right)}.$$This function shows a maximum at the value αT=0.75·π=2.35, i.e., if *T* is chosen for the length of 0.75·Q oscillations. This maximum is equal to 40% of the value we obtain by hard switching the stimulation on and off:(66)$${\left.u_{A,\;Hann}\right|}_{T\alpha =2.35}\approx 0,40\cdot {\left.u_{A,\;cut}\right|}_{T\alpha =\pi }.$$Figure 17 shows in blue the driving voltage and in black the corresponding response signal of the resonator specified in Figure 3 for a driving signal which is weighted in the time domain with a Hann window. The driving frequency was set to resonance frequency, at the 3 dB band edge and at twice the 3 dB band edge. The length of the Hann windows is optimized to maximize the response signal at the time when the driving signal is switched off.

## 6. Stimulating a Resonator by Using a Frequency-Modulated Driving Signal


In an up chirp, the instantaneous frequency varies linearly in the time interval $T_{Chirp}$ across the bandwidth $B_{Chirp}$ from the angular frequencies $\omega_{low}$ to $\omega_{high}$. The rate of change is called the chirp rate μ, with
(67)$$\mu =\frac{B_{Chirp}}{T_{Chirp}}=\frac{f_{low}-f_{high}}{T_{Chirp}}.$$In a down chirp, the instantaneous frequency varies linearly in from $f_{high}$ to $f_{low}$. The driving voltage $u_{0}\left(t\right)$ can be written as
(68)$$u_{0}\left(t\right)=U_{0}\;\sigma \left(t\right)\cdot \left\{\begin{array}{cc} e^{j\left(\omega_{low}+2\pi \mu t\right)\cdot t} & \mathrm{for}\;\mathrm{upchirp}\\ e^{j\left(\omega_{high}-2\pi \mu t\right)\cdot t} & \mathrm{for}\;\mathrm{downchirp} \end{array}\right\}$$We can write for the generated signal $u_{A}\left(t\right)$ in the source resistor of the antenna
$$u_{A}\left(t\right)=\int_{-\infty }^{\infty }h_{A}\left(t-\tau \right)\cdot u_{0}\left(\tau \right)\;d\tau .$$The following analysis is performed for an up chirp. The equations for a down chirp are corresponding. The analysis of above integral leads to two Fresnel integrals (Equation (A280)) which cannot be integrated analytically
(69)$$\begin{array}{ccc} u_{A}\left(t\right) & = & U_{0}\frac{\alpha_{A}}{\omega_{d}}\left[\left(\omega_{d}-j\alpha \right)e^{-\alpha t-j\omega_{d}t}\int_{0}^{t}e^{+\alpha \tau +j\left(\omega_{d}+\omega_{low}\right)\tau +j2\pi \mu \tau^{2}}d\tau \right.\\ & & \left.\;+\left(\omega_{d}+j\alpha \right)e^{-\alpha t+j\omega_{d}t}\int_{0}^{t}e^{+\alpha \tau -j\left(\omega_{d}-\omega_{low}\right)\tau +j2\pi \mu \tau^{2}}d\tau \right]. \end{array}$$We can solve this integral numerically or use an approximation method based on the so-called stationary phase method (see Appendix H). The quadratic terms in the exponents result in rapidly oscillating phase functions. Since the amplitudes remain constant, time domains with a rapidly oscillating phase do not contribute to the output of the integral. Only sections with a slowly varying phase contribute to the output of the integral. This is only the case if the chirp modulation μτ is close to the resonant frequency $+\omega_{d}$.

Figure 18 shows the contribution of the stimulation chirp signal to the oscillating signal in the resonator for the resonator characterized in Figure 3 and a chirp function over a length of T=400 with a relative bandwidth of 20% centered at center frequency $f_{0}$. The left graph shows the real part of the driving voltage with respect to the resonance frequency, and the right graph the real and imaginary part of the integral over this driving voltage as a function of *t*. Figure 18 illustrates this principle of the stationary phase. Only the part where the chirp modulation μτ hits the resonant frequency, which is the middle in the left figure, contributes significantly to the response signal. All other oscillations cancel each other out. This is also illustrated in the right diagram, where only this part leads to a significant contribution in the integral.

The chirp modulation reaches the angular natural frequency $\omega_{d}$ of the resonator at the time $\tau_{s}$, with
(70)$$\tau_{s}=\frac{\omega_{d}-\omega_{low}}{2\pi \mu }.$$Our approximation limits the integrals in Equation (69) to this stationary range, where the phase of the stimulating signal matches the phase of the excited oscillation to $\pm \frac{\pi }{2}$. This is the time interval (see Appendix H, Equations (A286) and (A287))
(71)$$\tau_{s}-\frac{1}{2\sqrt{\mu }}\leq \tau \leq \tau_{s}+\frac{1}{2\sqrt{\mu }}.$$The instantaneous frequency *f* of the chirp in this time interval sweeps between
(72)$$f_{0}-\frac{1}{2}\sqrt{\mu }\leq f\leq f_{0}+\frac{1}{2}\sqrt{\mu }.$$Within this stationary range, the frequency of the stimulation signal is set constant to the resonance frequency. With these approximations, we obtain for the response signal within the stationary range $\left(\tau_{s}-\frac{1}{2\sqrt{\mu }}\leq t\leq \tau_{s}+\frac{1}{2\sqrt{\mu }}\right)$, Equation (A296))
(73)$$u_{A}\left(t\right)\approx U_{0}\frac{\alpha_{A}}{\omega_{d}}\frac{\omega_{d}+j\alpha }{\alpha }\left(1-e^{-\alpha \left(t-\tau_{s}+\frac{1}{2\sqrt{\mu }}\right)}\right)e^{+j\omega_{d}t}.$$The chirp signal starts to stimulate the resonator at the beginning and stops stimulating at the end of this range. After the stimulation, the resonator decays with (Equation (A302))
(74)$$u_{A}\left(t\right)\approx U_{0}\frac{\alpha_{A}}{\omega_{d}}\frac{\omega_{d}+j\alpha }{\alpha }\left(1-e^{-\alpha \frac{1}{\sqrt{\mu }}}\right)e^{-\alpha \left(t-\tau_{s}-\frac{1}{2\sqrt{\mu }}\right)}e^{+j\omega_{d}t}\;\;\;\mathrm{for}\;\;\;t>\left(\tau_{s}+\frac{1}{2\sqrt{\mu }}\right).$$The stimulation the resonator with the chirp signal is limited by the time length of the stationary range $\tau_{s}$
(75)$$\tau_{s}=\sqrt{\frac{1}{\mu }}.$$The resonator will be fully loaded and, thus, the decay signal will be maximum if $\alpha \tau_{s}>\pi .$

A down chirp starts to load the resonator at a frequency of $f_{0}+0.5\sqrt{\mu }$ and stops loading at $f_{0}-0.5\sqrt{\mu }$. Readout systems using chirped signals often mix down the response signal with the transmit signal for signal detection. In this case, the maximum response signal will result at $f_{0}+0.5\sqrt{\mu }$ when using an up chirp and at $f_{0}-0.5\sqrt{\mu }$ when using a down chirp.

Figure 19 shows the driving voltage $u_{0}\left(t\right)$ with the dashed blue line and the response signal of the resonator specified in Figure 3. The full black line shows the response signal calculated analytically according to the approximation of stationary phase and the dotted black line shows the numerical simulation of the response signal using Matlab. The drive signals are modulated with a linear chirp of length 400. The chirp bandwidths and thus the chirp rate are varied, whereby in the left graph, the bandwidth is 10% of $f_{0}$, which results in an $\alpha \tau_{s}=2$. A chirp bandwidth of 20% and thus an $\alpha \tau_{s}=1.4$ was used for the middle and for the right graph, 40% bandwidth with an $\alpha \tau_{s}=1$ was used. The bandwidths of the chirps are centered around the center frequency of the resonator, the chirps reach the resonance frequency at time position 200. Due to the charging of the resonator in the synchronous range, the maxima of the responses occur at the end of the synchronous range. The instantaneous frequencies of the chirps are already more advanced at this point and are at $f_{0}\pm \sqrt{0.5\pi \mu }$, depending on the up or down chirp. In the numerical simulations with MATLAB, the maximum of the response signal is shifted slightly to later times due to the different wave forms. The red asterisks give the synchronous range and the blue crosses the maximum of the numerical calculated response.

## 7. Discussion and Summary

In this contribution, the external excitation of a resonator, which is wired to an antenna in a wireless passive sensor system, and the subsequent decay characteristic of the stored energy was modeled analytically and numerically, respectively. The resonator is modeled as a series RLC circuit, the external excitation is given by the readout signal of the wireless sensor system.

During stimulation, the resonator oscillates in a forced oscillation. At the beginning and with every change in the stimulation, additional natural oscillations are excited due to the boundary conditions, since the exciting CW signal is not a solution of the descriptive differential equation. The physical boundary conditions in the RLC equivalent circuit are the continuity of the current in the coil and the continuity of the capacitor voltage. Due to the boundary conditions, natural oscillations, damped cosine and sine oscillations, or $e^{\left(\alpha +j\omega_{d}\right)t}$ and $e^{\left(\alpha -j\omega_{d}\right)t}$ are always excited together. If the frequency of the exciting signal is close to the resonance frequency, then when excited with a cosine oscillation, the associated damped natural cosine oscillation will be several times the quality factor more strongly excited than the natural sine oscillation. The same applies to an excitation with a complex exponential function.

At the beginning of the excitation, the natural oscillations have the same amplitude, but with the opposite sign of the forced oscillation. As the natural oscillations gradually decay, more and more energy is stored in the resonator. If the exciting frequency and the frequency of the natural oscillation do not match, beats will occur during the transient process. The simultaneous occurrence of the natural oscillations and the forced oscillation characterizes the transient process, which comes to an end as the natural oscillations decay. Natural oscillations are also generated at the end of the excitation. These take over all the energy stored in the resonator and the forced oscillation ends immediately.

Resonators are linear, time-invariant systems that react with the same frequencies as they are excited. Any change in the exciting signal, such as switching on and off, are non-linear processes that will generate natural oscillations when adjusting to the new state. This generation of natural oscillations is also independent of whether the change in the stimulating signal is discontinuous, such as a hard switching on and off, or continuous and constantly differentiable, as when using a Tukey window.

The natural oscillations have the same temporal symmetry as the excitation signal. With hard switching on and off, the spectrum of the natural oscillation created by turning off the excitation is, therefore, a time-delayed and inverted copy of the spectrum of the natural oscillation that will arise when that excitation was turned on. The joint spectral power density of all natural oscillations was taken from the excitation spectrum. However, the individual spectrum of the natural oscillation that results from switching off an excitation can contain spectral components that were not part of the spectral power density of the excitation, provided that these spectral components are compensated for by the spectrum of the natural oscillation that was generated when this excitation was switched on.

The signal responses of the resonator to different temporal waveforms of readout signals were analyzed analytically and numerically. These signals included a rectangular, a trapezoidal, and a Tukey window CW signal as well as a frequency-modulated readout signal with a chirp function. The most efficient way to readout a wireless resonator is to use a CW signal with a rectangular, hard switched on and off waveform applied for Q oscillations. If the source resistance of the antenna is matched to the loss resistance of the resonator, the resonator in this case will sent back a decaying response signal, which begins at a power level of half the power that was delivered to the resonator by the antenna during excitation.

Readout signals weighted in the time domain with a trapezoidal or Tukey window require a lower bandwidth than a rectangular window signal. However, their response signals also show a lower level at the time the readout signal stops because the additional excitation in the falling edge does not fully compensate for the exponential drop in the response signal. A comparison of the amplitude of the decay signals generated by a readout signal with a trapezoidal window $u_{A,\;Tapezoidal}\left(t\right)$ or a Tukey window $u_{A,\;Tukey}\left(t\right)$ with the decay signal in the case of a hard switching on and off of the readout signal $u_{A,\;cut}\left(t\right)$ gives for a fast roll off ($\left|\left(\alpha -j\omega_{d}+j\omega \right)\left(T-T_{2}\right)\right|<1$):(76)$$\begin{array}{ccc} u_{A,\;Tapezoidal}\left(t\right) & \approx & u_{A,\;cut}\left(t\right)\left(1-\frac{1}{2}\left(\alpha -j\omega_{d}+j\omega \right)\left(T-T_{2}\right)\right.\\ & & \left.\;+\frac{1}{6}{\left(\alpha -j\omega_{d}+j\omega \right)}^{2}{\left(T-T_{2}\right)}^{2}\right) \end{array}$$
(77)$$\begin{array}{ccc} u_{A,\;Tukey}\left(t\right) & \approx & u_{A,\;cut}\left(t\right)\left(1-\frac{1}{2}\left(\alpha -j\omega_{d}+j\omega \right)\left(T-T_{2}\right)\right.\\ & & \left.\;+\left(\frac{1}{4}-\frac{1}{\pi^{2}}\right){\left(\alpha -j\omega_{d}+j\omega \right)}^{2}{\left(T-T_{2}\right)}^{2}\right) \end{array}$$

In this estimation, equal times with constant excitation were used for all three windows. We obtain similar decay signals with both windows however with a lower readout bandwidth. The term linear in $\left(T-T_{2}\right)$ and the quadratic term will not cancel, since this would require $\left(\alpha -j\omega_{d}+j\omega \right)\left(T-T_{2}\right)\approx 3$, where the small term approximation of the exponential function is no more valid.

Time domain-weighted readout signals without any flat-top component, such as Hann or triangle-weighted signals, can also be used to readout wireless resonators if their time-domain length is matched to the quality factor Q of the resonator. Signals without flat-top components typically require quite a small bandwidth. The amplitude of the decay signals generated by a readout signal with a triangular or Hann window with a time length of 0.80·Q or 0.75·Q oscillations reaches 40% (−8 dB) of the amplitude of the decay signal generated by hard switching on and off.

Figure 20 shows a comparison of the examined excitation signals. The schematic representation of the waveforms in the time and frequency domain uses the same scaling for all window functions. The data given in the table were calculated using the framework data presented in Section 2.4 and the stimulation functions introduced in Section 3, Section 4 and Section 5. The length of the constant stimulation plateaus for the rectangular, trapezoidal, and Tukey windows was set to $Q/f_{0}$, the resonator then oscillates with 97% of the maximum amplitude. For the trapezoidal and Tukey windows, additional 10% of this length was used for the rising edge and 10% for the falling edge. The roll-off characterizes the reduction in the response signal at the end of the excitation signal compared to a full charge of the resonator. The rectangular excitation signal provides the highest response signal but at the expense of a fairly large signal bandwidth. A trapezoidal or cosine-weighted excitation signal requires significantly less bandwidth at the expense of a 1 dB lower response signal and a slight increase in the duration of the interrogation signal. The triangular and Hann windows are more compact in both the time and frequency domain but at the expense of a 6 dB lower response signal.

If a chirp with the bandwidth $B_{Chirp}$ and the length $T_{Chirp}$ is used as readout signal, it will essentially only excite the resonator during the time in which there is synchronization between the chirp signal and the natural oscillation of the resonator. The duration of this synchronization is $\sqrt{T_{Chirp}/B_{Chirp}}$. The strength of the response signal results from the stimulation of the resonator during this length of time. The maximum of the response signal will not occur at the time when the chirp signal matches the resonant frequency, but is delayed by $\pm 0.5\sqrt{T_{Chirp}/B_{Chirp}}$ depending on whether an up or down chirp was used.

## Figures and Tables

**Figure 1 sensors-24-01323-f001:**
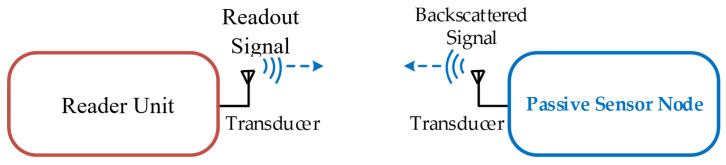
Schematic of a wireless passive sensor system of delay line or resonator type: A reader sends out a readout signal over a transducer into a wireless channel to a passive sensor node. There, this signal is picked up with the help of a second transducer and then stored in a delay line or in a high Q resonator. When the readout signal is turned off, a part of the signal, which has been stored as an excitation in the passive battery- and IC-free sensor node, will be sent out as backscattered signal, which is picked up by the reader unit and evaluated.

**Figure 2 sensors-24-01323-f002:**
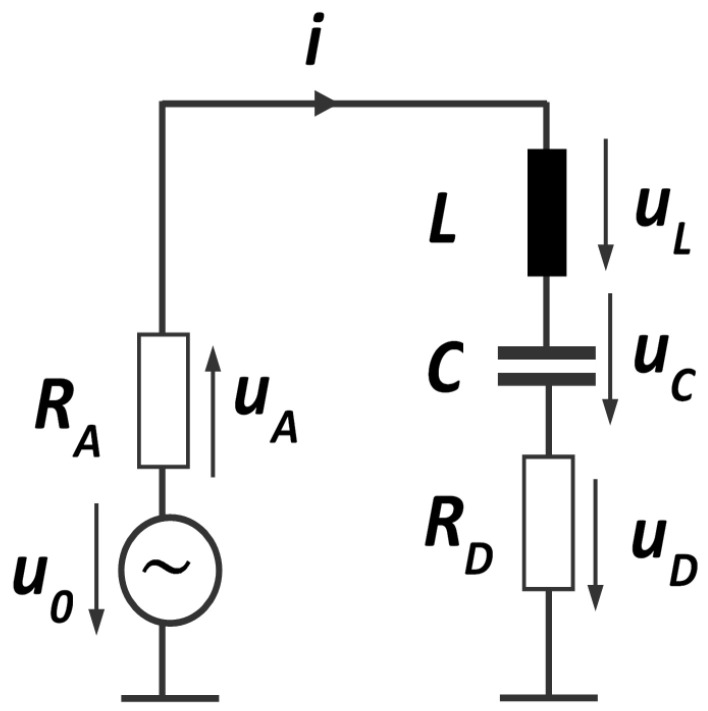
Electrical circuit for analyzing a wireless readout resonator. The antenna is modeled by a voltage source $u_{0}$ with internal resistance $R_{A}$ and the resonator with a series circuit of *L*, *C*, and $R_{D}$.

**Figure 3 sensors-24-01323-f003:**
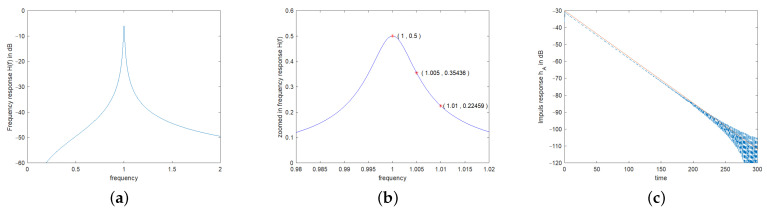
(**a**) Frequency response $H_{A}\left(f\right)$ according to Equation (Equation 14) in dB, (**b**) zoomed-in frequency response in linear scale, and (**c**) impulse response $h_{A}\left(t\right)$ according to Equation (Equation 15) of an example resonator. For the example resonator, the center frequency was set to 1 and the quality factor *Q* to 100. Electrical matching was applied ($R_{A}=R_{D}$). In graph (**b**), 3 markers were placed, (i) at center frequency, (ii) at the upper 3 dB band edge and (iii) at twice this frequency spacing from center frequency. The impulse response given in (**c**) shows in red (full line) Equation (Equation 15) and in blue (dashed line) the IFFT from $H_{A}\left(f\right)$ calculated with MATLAB [29], with the latter being shifted downwards by 1 dB to become visible. The impulse response would show heavy oscillations due to the 2 contributions at $\pm \omega_{d}$. To suppress these oscillations, only one part corresponding to $+\omega_{d}$ was used in the formula and also for Matlab. In the impulse response calculated with MATLAB we see aliasing since no weighting was applied in the frequency response before inverse Fourier transforming.

**Figure 4 sensors-24-01323-f004:**
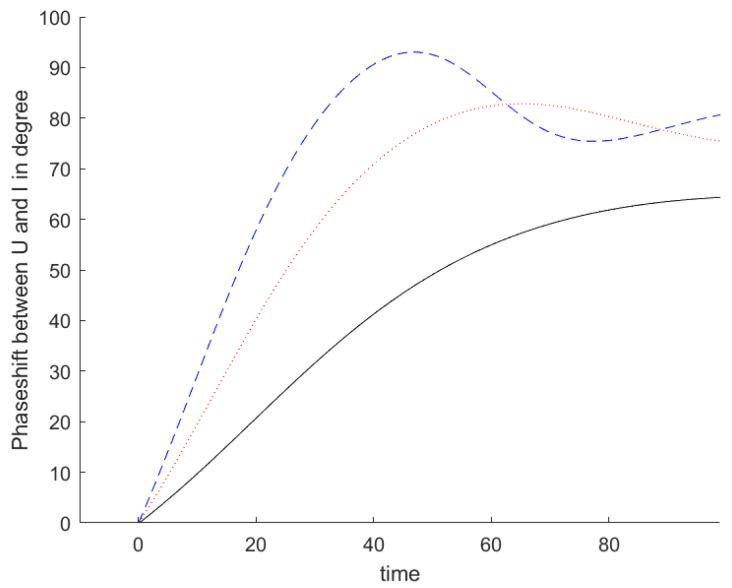
Phase shift in degrees between the voltage applied to the resonator and the current in the resonator for three frequencies below the resonance frequency as a function of the time since the start of the stimulation. The time is measured in units of the inverse resonance frequency. The solid black line shows the phase shift for a frequency at the lower 3 dB frequency while the red dotted line and dashed blue line show the phase shifts at two and three times this frequency offset from the resonance frequency, respectively.

**Figure 5 sensors-24-01323-f005:**
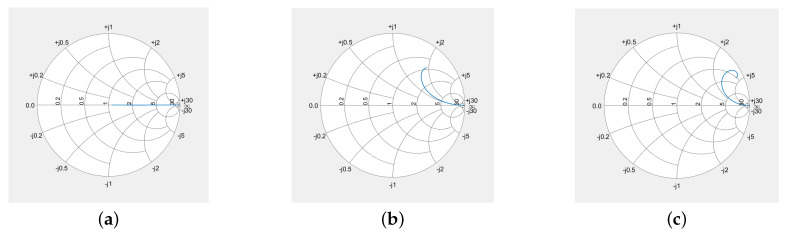
Time-dependent reflection coefficient $S_{11}$ at the port between antenna and resonator, (**a**) for resonance frequency, (**b**) for a frequency at the 3 dB corner, and (**c**) at a frequency twice this distance from the resonance. After switching off the stimulation, the voltage and current is in phase, the resonator, however, now acts as source.

**Figure 6 sensors-24-01323-f006:**
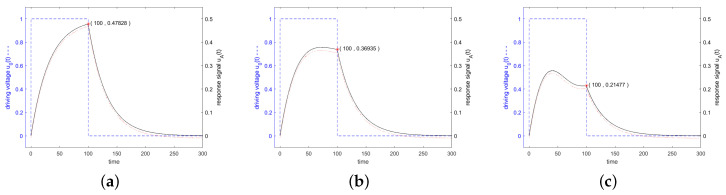
Driving voltage $u_{0}\left(t\right)$ in blue dashed line and response signal in black solid line of the resonator specified in Figure 3. The stimulating frequency was set to $f=f_{0}$ in (**a**), $f=0.995f_{0}$ in (**b**), and $f=0.99f_{0}$ in (**c**). The stimulating signal shows a rectangular envelope in time domain and lasts *Q* oscillations, which start at t=0. The response signals are calculated according to Equations (Equation 22) and (Equation 27) and are shown at a scale enlarged by a factor of 2 when compared to the scale of the driving voltage. The red dotted line shows the result of a numerical calculation with MATLAB, shifted down by 0.01 to become visible. A red marker was set in each graph at the end of the driving interval.

**Figure 7 sensors-24-01323-f007:**
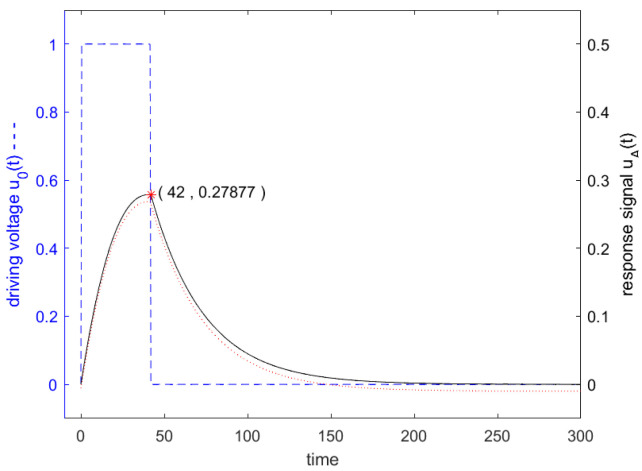
Driving voltage $u_{0}\left(t\right)$ in blue dashed line and response signal in black solid line of the resonator specified in Figure 3. The driving signal shows a rectangular envelope in the time domain and lasts 0.42Q oscillations, which start at t=0. The driving frequency is the same as in Figure 6 right graph, $f=0.99f_{0}$.

**Figure 9 sensors-24-01323-f009:**
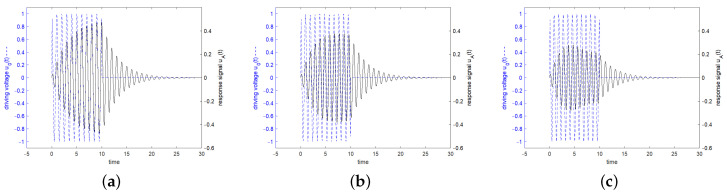
Real part of the driving voltage $u_{0}\left(t\right)$ in blue dashed line and real part of the response signal in black solid line of the resonator specified in Figure 3, however for visualization with a quality factor of 10. The driving frequency was set to $f=f_{0}$ in (**a**), the left 3 dB band edge in (**b**), and twice the left 3 dB band edge in (**c**). The driving signal shows a rectangular envelope in time domain and lasts *Q* oscillations, which start at t=0. The frequency of the response signal approaches the frequency of the forced oscillation in the driven interval. The phase difference between the driving signal and the response signal in the case that the exciting frequency is not equal to the angular natural frequency of the resonator can also be seen. After switching off the driving voltage, the resonator oscillates at the natural angular frequency.

**Figure 10 sensors-24-01323-f010:**
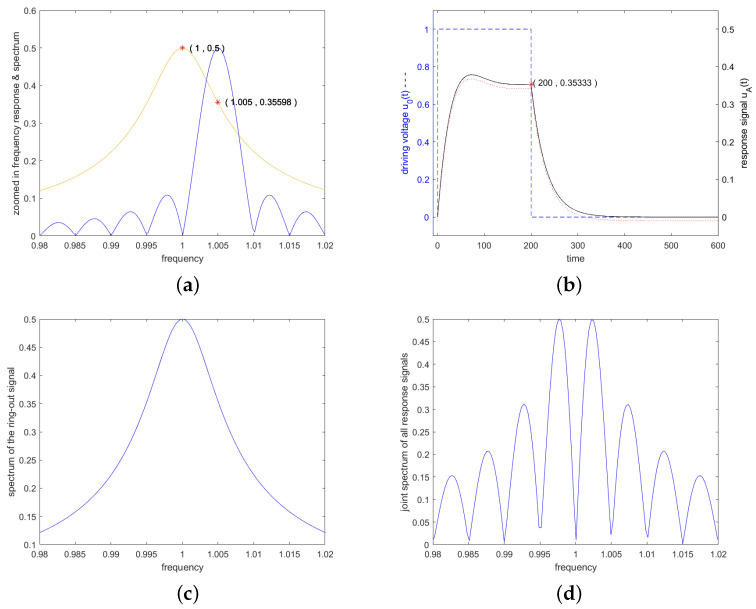
Generation of a response signal at a frequency that was not included in the excitation spectrum: Figure (**a**) shows the Lorentz curve of the resonator in red and in blue the spectrum of an excitation signal at a carrier frequency of $f_{0}\cdot \left(1+0.5/Q\right)$ for a length in time domain of $Q/f_{0}$. Red marker was placed on the Lorentz curve at center frequency and at the 3 dB point. Figure (**b**) shows the response signal of the resonator to this excitation signal, i.e., the forced oscillation and the subsequent decay signal. A red mark is placed at the end of the forced oscillation. Figure (**c**) shows the spectrum of the decay signals alone and Figure (**d**) shows the combined spectrum of the response signals from the beginning and the end of the excitation. All other details corresponded to the resonator shown in Figure 3.

**Figure 11 sensors-24-01323-f011:**
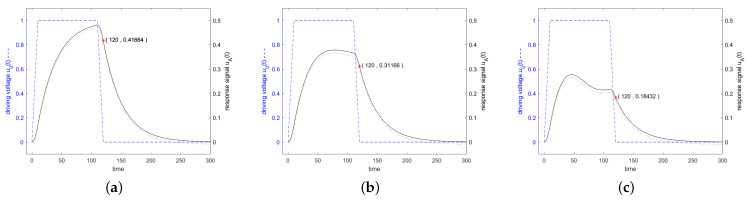
Driving voltage $u_{0}\left(t\right)$ and response signal of the resonator specified in Figure 3. In Figure (**a**), the stimulating frequency was set to $f=f_{0}$, in (**b**) to $f=0.995f_{0}$, and in (**c**) to $f=0.99f_{0}$. The stimulating signal shows a trapezoidal envelope in the time domain with a length in the constant range of $Q/f_{0}$. The linearly increasing and decreasing parts are $0.1\cdot Q/f_{0}$. A red marker was set in each graph at the end of the driving interval. All other details are the same as for Figure 6.

**Figure 12 sensors-24-01323-f012:**
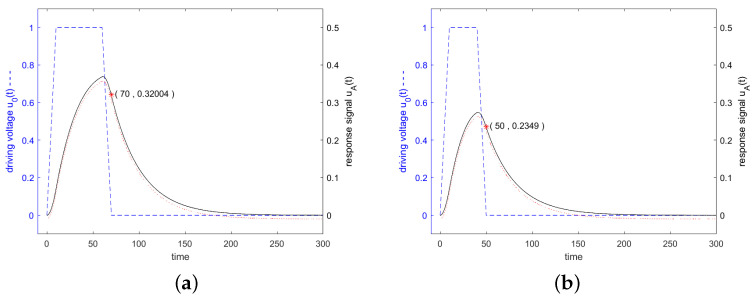
Driving voltage $u_{0}\left(t\right)$ and response signal of the resonator specified in Figure 3 and Figure 7. The driving signal was shortened in time domain, when compared to the settings for Figure 11, to maximize the response signal. Figure (**a**) shows the response signal for a driving frequency at the upper 3 dB band edge, $f=0.995f_{0}$ (left graph), but here the length in the constant range was shortened to $0.5\cdot Q/f_{0}$, and keeping the linearly increasing and decreasing parts at $0.1\cdot Q/f_{0}$. Figure (**b**) shows the response signal for a driving frequency of $f=0.99f_{0}$ and a shortened constant range of $0.3\cdot Q/f_{0}$. A red marker was set in each graph at the end of the driving interval.

**Figure 13 sensors-24-01323-f013:**
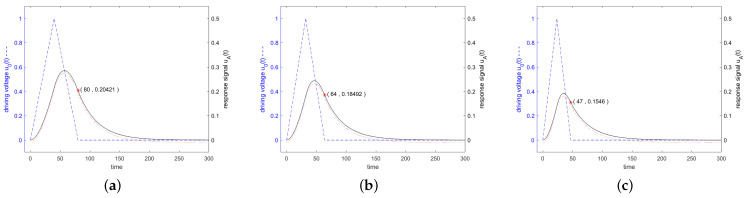
Driving voltage $u_{0}\left(t\right)$ and response signal of the resonator specified in Figure 3. The envelope of the driving signal is weighted in the time domain with a triangle function (Bartlett window). The driving frequency was set to $f=f_{0}$ in Figure (**a**), $f=0.995f_{0}$ in (**b**), and $f=0.99f_{0}$ in (**c**). The length of the Bartlett window was optimized of maximum response signal after switching off the driving signal. A red marker was set in each graph at the end of the driving interval. All other details are the same as in Figure 6.

**Figure 14 sensors-24-01323-f014:**
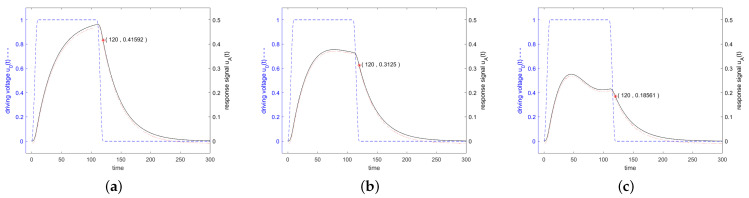
Driving voltage $u_{0}\left(t\right)$ and response signal of the resonator specified in Figure 3. The driving signal is weighted in the time domain according to a Tukey window with a length of $Q/f_{0}$ in the constant range. The cosine increasing and decreasing parts are $0.1\cdot Q/f_{0}$. The driving frequency was set to $f=f_{0}$ in Figure (**a**), to $f=0.995f_{0}$ in (**b**), and to $f=0.99f_{0}$ in (**c**). A red marker was set in each graph at the end of the driving interval. All other details are the same as for Figure 6.

**Figure 15 sensors-24-01323-f015:**
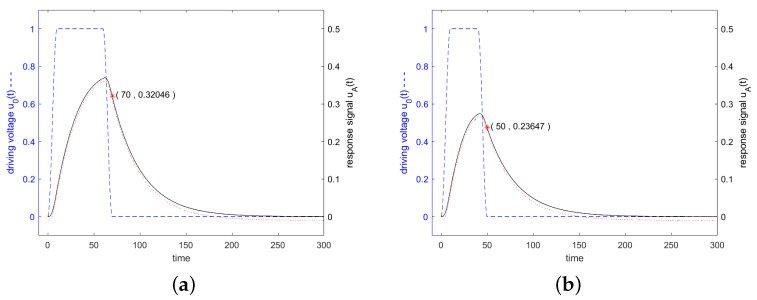
Driving voltage $u_{0}\left(t\right)$ and response signal of the resonator specified in Figure 3. The driving signal was shortened in the time domain when compared to the settings for Figure 14, to maximize the response signal. Figure (**a**) shows the response of the resonator for a stimulating frequency at the upper 3 dB band edge, $f=0.995f_{0}$, where the length in the constant region was shortened to $0.5\cdot Q/f_{0}$, but the rising and falling cosine parts of length $0.1\cdot Q/f_{0}$ were retained. Figure (**b**) shows the response signal for a driving frequency of $f=0.99f_{0}$ and a shortened constant range of $0.3\cdot Q/f_{0}$. A red marker was set in each graph at the end of the driving interval.

**Figure 16 sensors-24-01323-f016:**
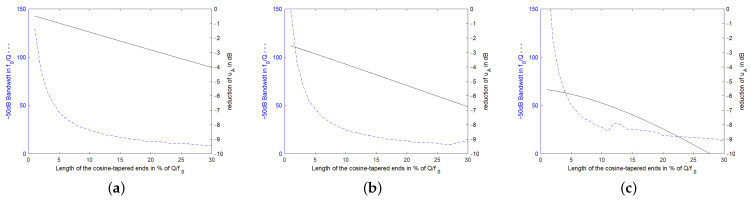
Reduction in the required bandwidth and corresponding decrease in the decaying response signal of the resonator specified in Figure 3 by appending a cosine-weighted edge as a function of the length of the cosine weighting. The resonator was stimulated in Figure (**a**) at resonance frequency, $f_{0}$, for a length in time domain of $Q/f_{0}$ in the constant stimulation range. In Figure (**b**), the stimulation signal is set to $0.995\cdot f_{0}$ for a time of $0.75\cdot Q/f_{0}$ in the constant range, and in (**c**) the stimulation is at $0.99\cdot f_{0}$ for $0.3\cdot Q/f_{0}$ in the constant range.

**Figure 17 sensors-24-01323-f017:**
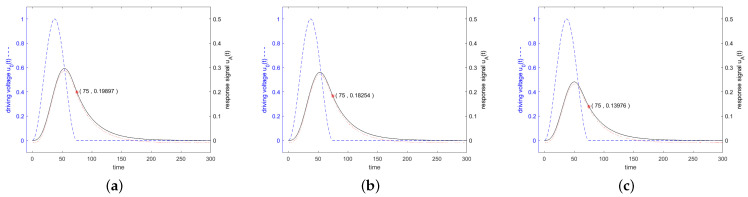
Driving voltage $u_{0}\left(t\right)$ and response signal of the resonator specified in Figure 3. The driving signal is weighted according to a Hann window with a time length to maximize the response signal. The driving frequency is in Figure (**a**) at resonance, in Figure (**b**) at the upper 3 dB band edge and twice this frequency distance as for (**c**).

**Figure 18 sensors-24-01323-f018:**
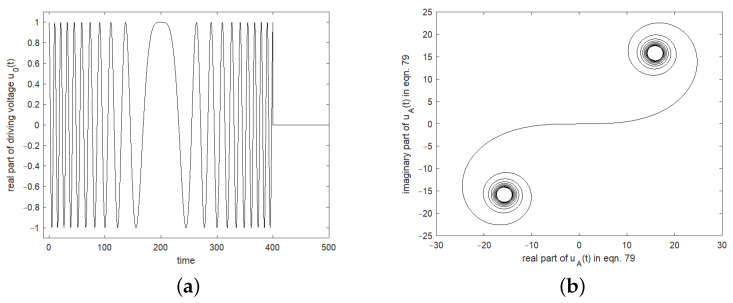
Figure (**a**) shows the real part of the driving voltage generated by a stimulating chirp signal over a relative bandwidth of 20% centered at resonance frequency $f_{0}$ and a time length of 400. Figure (**b**) shows the integral over this driving voltage, the so-called Euler spiral or Cornu spiral. Only the center, stationary part around the resonance frequency contributes to the resonant oscillation, all other parts cancel each other out.

**Figure 19 sensors-24-01323-f019:**
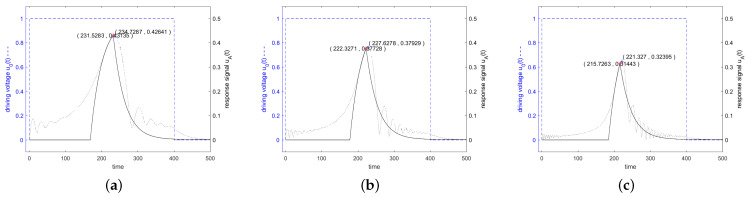
Driving voltage $u_{0}\left(t\right)$ (in dashed blue line) and response signal (in black line) of the resonator specified in Figure 3. The driving signal is modulated using a linear chirp with a chirp rate resulting in $\alpha \tau_{s}$ of 2 in Figure (**a**), 1.4 in (**b**), and 1 in (**c**). The full black line shows the response signal calculated analytically according to the approximation of stationary phase and the dotted black line shows the numerical simulation of the response signal using MATLAB. The red asterisk gives the end of the synchronous range and the blue cross the maximum of the numerical calculated response.

**Figure 20 sensors-24-01323-f020:**
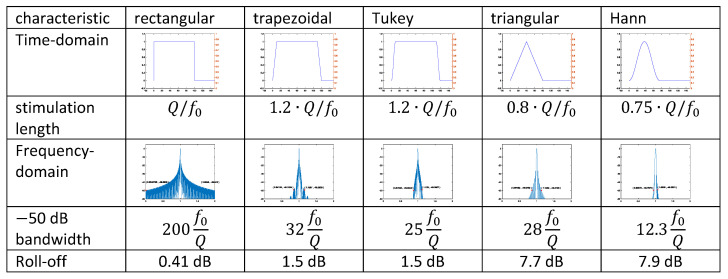
Comparison between studied excitation signals.

## Data Availability

The data presented in this study are available on request from the corresponding author.

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
