# Peer review of "Optimal Excitation and Readout of Resonators Used as Wireless Passive Sensors"

_sensors, 2024, doi:10.3390/s24041323_

Round 1

Reviewer 1 Report

Comments and Suggestions for Authors

This paper reports the modeling of excitation and readout of resonators as wireless passive sensors. Various conditions including natural oscillation without excitation, steady state excitation, decay characteristics, switch on, switch off, increasing and decreasing driving voltage, frequency modulated driving signal, etc. are explored. Both analytical results derived and numerical simulations through MATLAB are presented and compared. It is valuable to have this compressive modeling of wireless passive sensors with resonators, which have wide applications.

However, I am confused about the analytical and simulation methods in the paper. What are the equations in the MATLAB simulation for the passive sensors with resonators? What are the differences between these equations and the ones derived in this paper? What are the contributions of the paper if these equations are the same or similar?

At the same time, please label the sub-figures in each figure and give descriptions accordingly.

Thank you.

Author Response

The authors would like to thank the reviewers for their valuable time and advice, which significantly improved the manuscript. The manuscript was revised according to their suggestions. What was done in detail:

Comments and Suggestions for Authors

This paper reports the modeling of excitation and readout of resonators as wireless passive sensors. Various conditions including natural oscillation without excitation, steady state excitation, decay characteristics, switch on, switch off, increasing and decreasing driving voltage, frequency modulated driving signal, etc. are explored. Both analytical results derived and numerical simulations through MATLAB are presented and compared. It is valuable to have this compressive modeling of wireless passive sensors with resonators, which have wide applications.

However, I am confused about the analytical and simulation methods in the paper. What are the equations in the MATLAB simulation for the passive sensors with resonators? What are the differences between these equations and the ones derived in this paper? What are the contributions of the paper if these equations are the same or similar?

Answer of the authors:

The same formulas and signals were used in MATLAB as in the analytical calculation. For the analytical calculations, the convolution of the excitation signals with the impulse response was calculated analytically; for chapter 3 “Switching the readout signal on and off”, the relevant differential equation was also solved directly (see appendix C). For the MATLAB results, the convolution was calculated numerically by Fourier transforming the excitation signals, multiplying them by the transfer function of the resonator and transforming them back. The curves from the numerically calculated results lie indistinguishably on the analytically calculated curves in all graphics. In order to make them visible, the numerical calculated curves were shifted downwards by the value 0.01 and drawn in red dots in the x-y graphics.

The compilation of the analytically and numerically calculated results shows first of all the correctness of the analytical formula, as it has been independently verified. The numerically generated results can be obtained much more quickly, the analytically derived results allow further investigations, as shown in the manuscript, as well as further optimizations, limit value considerations, etc.

The subsection 2.4 “Numerical Analysis”, which is now “Analytical and numerical analysis” was re-written, to explain the simulation in more detail.

At the same time, please label the sub-figures in each figure and give descriptions accordingly.

Answer of the authors: Done.

Reviewer 2 Report

Comments and Suggestions for Authors

In this paper, the authors model the external excitation of a resonator with different excitation signals and its subsequent decay characteristics. The response corresponds to a rectangular, a trapezoidal, a Tukey window are given analytically and numerically. I think it is valuable for the use of resonators in wireless passive sensors. But there are still some questions to consider.

1. The paper is too long to read. Note that some of the contents are well known, such as sections 2.1 and 2.2, etc., could you abbreviate them?

2. At the end of the article, the discussion and summary gives the readers an impression that the article is not over yet. Can the authors give a concise conclusion about their contribution? For the linear systems, the theory is mature, and a large number of deductions and calculations carried out by the author do not have much theoretical difficulty. I think the authors should highlight some new findings in his calculation.

3. The authors use red dotted lines to represent the numerical results, changes the value of the results, and there is no legend in the figures, which is difficult for readers.

Author Response

The authors would like to thank the reviewers for their valuable time and advice, which significantly improved the manuscript. The manuscript was revised according to their suggestions. What was done in detail:

Reviewer 2
Comments and Suggestions for Authors
In this paper, the authors model the external excitation of a resonator with different excitation signals and its subsequent decay characteristics. The response corresponds to a rectangular, a trapezoidal, a Tukey window are given analytically and numerically. I think it is valuable for the use of resonators in wireless passive sensors. But there are still some questions to consider.

  1. The paper is too long to read. Note that some of the contents are well known, such as sections 2.1 and 2.2, etc., could you abbreviate them?

Answer of the authors:
We have shortened sections 2.1, 2.2, etc, primarily by omitting commonly known properties. A number of texts have been streamlined. Sections 2.3 and2.4 have been combined. Unfortunately, the space gained was eaten up again by the more detailed description of the simulation in Chapter 2.4. All other chapters have also been shortened, in particular the elongated equations have been reduced in size by introducing a sum over both natural oscillations. The text was streamlined where possible. The manuscript was shortened from 40 pages to 35.

  1. At the end of the article, the discussion and summary gives the readers an impression that the article is not over yet. Can the authors give a concise conclusion about their contribution? For the linear systems, the theory is mature, and a large number of deductions and calculations carried out by the author do not have much theoretical difficulty. I think the authors should highlight some new findings in his calculation.

Answer of the authors: 
The abstract and discussion have been modified to avoid the impression that the article is ongoing and to highlight the new findings of the calculations. (In fact, the reviewer's impression is partly correct, as I am planning a next article applying the formulas.)

  1. The authors use red dotted lines to represent the numerical results, changes the value of the results, and there is no legend in the figures, which is difficult for readers.

Answer of the authors: 
Thank you for this suggestion. We see the problem. The red dotted lines for the results of the numerical simulation were, therefore, discussed again in Section 2.4. Without the downward shift, the lines would lie indistinguishably on the analytically calculated curves. If the reviewers recommend that these curves be highlighted more, we will change them. What do you suggest?

Reviewer 3 Report

Comments and Suggestions for Authors

In this paper, the authors used both analytical and numerical modeling to simulate the external excitation of a resonator with various excitation signals and its subsequent decay characteristics. However, some minor comments need to be addressed.

1.      The inference of analytical modelling, effectiveness and usefulness should be included in the abstract to add some value of the modelling.

2.      The reference is missing in line 110. i.e., how the general solution is obtained

3.      Will there be a difference in stimulation done in time-domain or by using the convolution with the impulse response (line 208).

4.      The quality of all figures can be improved.

5.      In figure 6, it is plotted versus time. Will this be same for all PC configurations. If it is configuration dependent have you mentioned your PC configuration?

6.      The inference of analysis using various windowing techniques can be detailed in summary

7.      Detail about the simulation length given in table 1 

Author Response

The authors would like to thank the reviewers for their valuable time and advice, which significantly improved the manuscript. The manuscript was revised according to their suggestions. What was done in detail:
Reviewer 3
In this paper, the authors used both analytical and numerical modeling to simulate the external excitation of a resonator with various excitation signals and its subsequent decay characteristics. However, some minor comments need to be addressed.
1. The inference of analytical modelling, effectiveness and usefulness should be included in the abstract to add some value of the modelling.
Answer of the authors: Done: Included in the abstract and in the summary
2. The reference is missing in line 110. i.e., how the general solution is obtained
Answer of the authors: Done
3. Will there be a difference in stimulation done in time-domain or by using the convolution with the impulse response (line 208).
Answer of the authors: Thank you for this valuable tip. No, there is no difference. In order to explain the simulation more clearly, section 2.4 has been rewritten.
4. The quality of all figures can be improved.
Answer of the authors: A label was added to all sub-figures and addressed in the text and in the image captions
5. In figure 6, it is plotted versus time. Will this be same for all PC configurations. If it is configuration dependent have you mentioned your PC configuration?
Answer of the authors: It is important that the frame data for the numerical simulation is chosen to be sufficiently large in both the time domain and the frequency domain so that aliasing is avoided. A corresponding sentence has been added in section 2.4 numerical analysis.
6. The inference of analysis using various windowing techniques can be detailed in summary
Answer of the authors: Appropriate sentences were added to the abstract and summary
7. Detail about the simulation length given in table 1 Answer of the authors: Done

Round 2

Reviewer 2 Report

Comments and Suggestions for Authors

All my comments are well responsed. I think it can be accepted in present form.